# Noise correlations and neuronal diversity may limit the utility of winner-take-all readout in a pop out visual search task

Ori Hendler[1,2]*, Ronen Segev[2,3,4], Maoz Shamir[1,2,5]

1 Department of Physics, Ben-Gurion University of the Negev, Beer-Sheva, Israel, 2 School of Brain Sciences and Cognition, Ben-Gurion University of the Negev, Beer-Sheva, Israel, 3 Department of Life Sciences, Ben-Gurion University of the Negev, Beer-Sheva, Israel, 4 Department of Biomedical Engineering, Ben-Gurion University of the Negev, Beer-Sheva, Israel, 5 Department of Physiology and Cell Biology, Ben-Gurion University of the Negev, Beer-Sheva, Israel

* hendlero@post.bgu.ac.il

## Abstract

Visual search involves active scanning of the environment to locate objects of interest against a background of irrelevant distractors. One widely accepted theory posits that pop out visual search is computed by a winner-take-all (WTA) competition between contextually modulated cells that form a saliency map. However, previous studies have shown that the ability of WTA mechanisms to accumulate information from large populations of neurons is limited, thus raising the question of whether WTA can underlie pop out visual search. To address this question, we conducted a modeling study to investigate how accurately the WTA mechanism can detect the deviant stimulus in a pop out task. We analyzed two types of WTA readout mechanisms: single-best-cell WTA, where the decision is made based on a single winning cell, and a generalized population-based WTA, where the decision is based on the winning population of similarly tuned cells. Our results show that neither WTA mechanism can account for the high accuracy found in behavioral experiments. The inherent neuronal heterogeneity prevents the single-best-cell WTA from accumulating information even from large populations, whereas the accuracy of the generalized population-based WTA algorithm is negatively affected by the widely reported noise correlations. These findings underscore the need to revisit the key assumptions explored in our theoretical analysis, particularly concerning the decoding mechanism and the statistical properties of neuronal population responses to pop out stimuli. The analysis identifies specific response statistics that require further empirical characterization to accurately predict WTA performance in biologically plausible models of visual pop out detection.

**Data availability statement:** The source code and all of the data used to produce the results and analyses presented in this manuscript are available on a GitHub repository at https://github.com/OriHendler/Winner-take-all-fails-to-account-for-pop-out-accuracy. We have also used Zenodo to assign a DOI to the repository: https://doi.org/10.5281/zenodo.13202915

**Funding:** This work has been supported in part by the Israel Science Foundation (grants no. 824/21 to RS; 624/22 to MS). The funders had no role in study design, data collection and analysis, decision to publish, or preparation of the manuscript.

**Competing interests:** The authors have declared that no competing interests exist.

## Author summary

Visual search is an important cognitive process that allows organisms to locate objects of interest within complex environments. Whether scanning a crowded scene or locating a specific item, the brain's ability to prioritize certain stimuli is essential for effective perception and decision-making. One widely accepted theory suggests that this process is governed by a winner-take-all algorithm, where the most salient stimulus suppresses competing signals to capture attention. This hypothesis has been supported by empirical studies and provides an elegant explanation for how the brain achieves saliency-based selection.

Here, however, we demonstrate that the winner-take-all algorithm cannot account for the high accuracy observed in pop out tasks. By combining a theoretical analysis and computational modeling, we reveal limitations in the winner take all framework and identify key factors that are likely missing in current understandings. These findings should encourage further exploration into the neural and computational mechanisms that enable the brain's exceptional capacity for saliency detection.

## Introduction

The primary aim of visual search is to locate a specific object within a cluttered visual environment. Ensuring an organism's survival demands both accuracy and speed, whether to detect food sources or pinpoint potential predators. In a visual search task, the object the observer is searching for is termed the target, and the non-target items are termed distractors. Humans and other vertebrates perform different visual search tasks at differing degrees of efficiency, which usually depend on differences between the target and the distractors [1–5]. There is a general consensus that there are two major search modes, known as parallel or pop out search, and serial search [5–13]. These two search modes have been reported in humans, monkeys, archer-fish, cats, and barn owls, thus illustrating the wide distribution of this visual behavior across vertebrate families [14–22].

 The distinction between these two search modes is perhaps best illustrated in a classic experiment designed to assess search task efficiency, where observers perform numerous search trials for an object while the number of distractors is varied. The time needed to locate the object; i.e., the reaction time, as well as the accuracy of the response are both measured.

 Most studies examining the pop out search mode report that differences in visual features between the target and the distractors help make the target more salient (Fig 1A-B) and lead to detection times that are independent of the number of distracting objects, as though the entire visual field were being processed in parallel (Fig 1D). This rapid response time is associated with very high accuracy: on a range of pop out tasks, humans commonly achieve success rates >96% [9,23], with only a slight decrease as the number of distractors is increased (Fig 1E) [24]. By contrast, in the

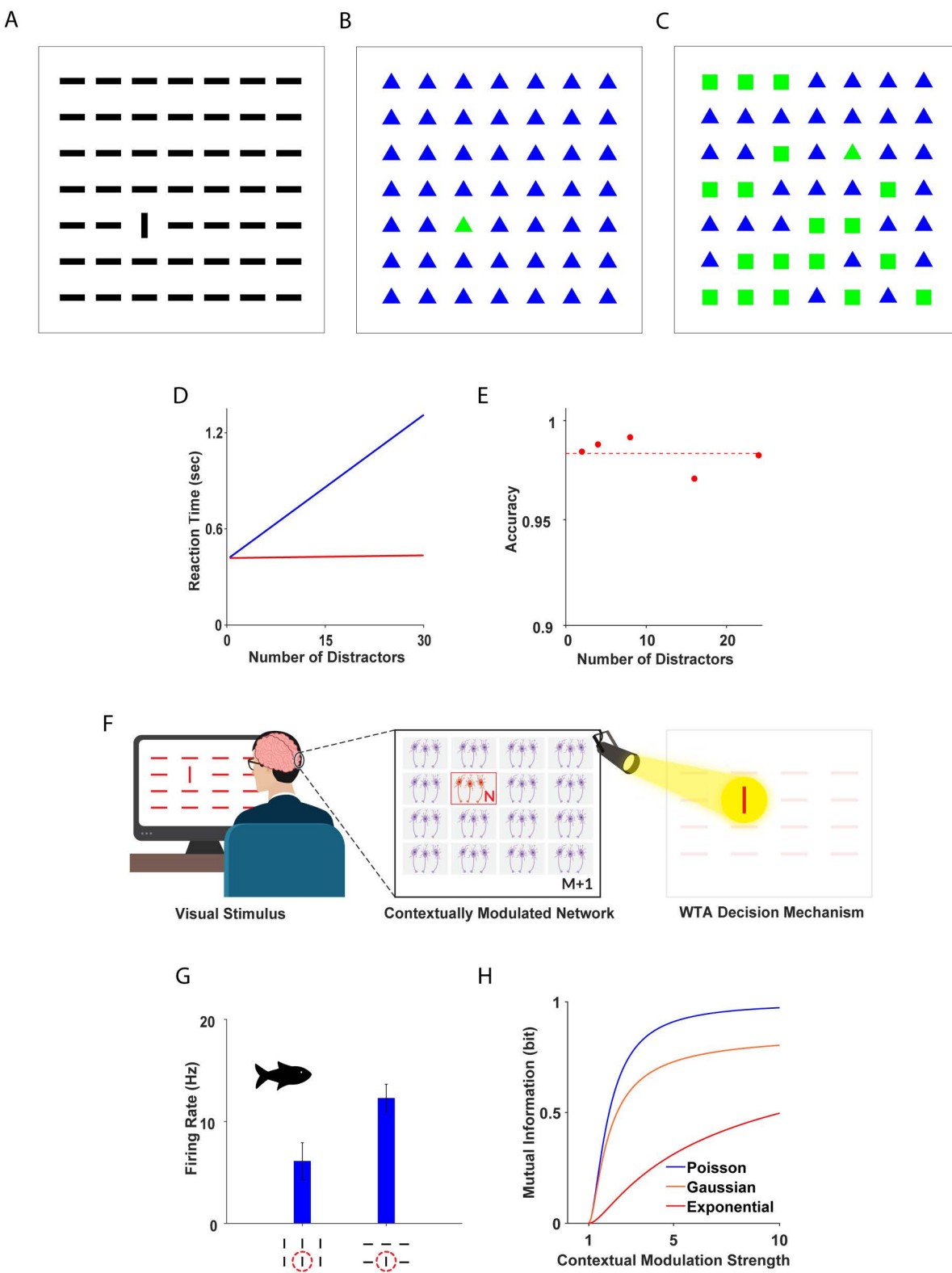

**Fig 1. Pop out: Behavior and Physiological Correlates. (A-B)** Illustration of two pop out stimuli: **(A)** A deviant vertical bar among numerous horizontal bars. **(B)** A deviant green triangle among many blue triangles. **(C)** A deviant green triangle among numerous green squares and blue triangles. **(D)**

Illustration of reaction times in serial (blue) and parallel (red) visual search tasks. **(E)** Accuracy of human subjects on a pop out task is plotted as a function of the number of distractors, adapted with permission from [24]. **(F)** Schematic illustration of the model system. From left to right: Pop out stimulus presented to the subject, system of $M+1$ populations of $N$ contextually modulated neurons each, WTA decision mechanism. **(G)** Example: mean firing rate of a single contextually modulated neuron in response to uniform and pop out stimuli from the optic tectum of a fish. Data courtesy of the Segev lab. The methodology and experimental procedure used to harvest these data are detailed in [21]. The dashed red line schematically depicts the classical receptive field of the neuron. **(H)** Mutual information of a neuron (see Methods) is presented as a function of its contextual modulation strength, $q$, for Poisson (blue), Gaussian (orange) and exponential (red) noise.

serial search mode, the target/distractor differences are less salient (Fig 1C), and no pop out is observed. In this case, the reaction time increases with the number of distractors (usually linearly, Fig 1D), and observers tend to perform serial visual scanning of the scene until the target is detected. Here we focus on the pop out search mode and in particular on its high accuracy.

The remarkable efficiency of pop out has prompted numerous experimental efforts to understand its underlying neural mechanism [20,25–32]. One widely accepted theory hypothesizes that pop out computation is implemented by a winner-take-all (WTA) competition between contextually modulated cells [6,33]. The basic model is shown in Fig 1F, which illustrates a task that consists of identifying the single deviant (vertical bar) from out of the set of (horizontal bar) distractors.

It is assumed that contextually modulated neurons respond to the pop out stimulus. These neurons are sensitive to objects placed outside their classical receptive field; i.e., they respond to the context of the stimulus. For example, data recorded from the archerfish optic tectum show that these cells fire at a higher rate when the stimulus within their classical receptive field is the deviant object than when the stimulus within their classical receptive field is a distractor (Fig 1G). Qualitatively similar results have been reported in the cat [34]. Contextually modulated neurons have been found in the visual systems of primates, cats, birds, and fish [21,32,34–37].

In the next stage of processing, the WTA algorithm estimates the location of the pop out object from the location of the receptive field of the single most active neuron. The WTA is a natural choice for selecting a single focus of attention or for identifying the single deviant stimulus in a pop out task. Thus, it has been widely assumed that a WTA mechanism is an essential part of the visual circuit responsible for directing attention [30,31,33,38–40].

Considerable effort has been devoted to investigating the implementation and dynamics of the WTA mechanism [41–45]. However, the ability of the WTA to correctly identify the winner has not received much attention. The accuracy of WTA mechanisms has been studied in the framework of the two-alternative forced choice task [46]. The findings indicate that in a competition between two homogeneous populations, WTA accuracy increases slowly with the number of neurons in each population. The accuracy of a temporal variant of the WTA that estimates the stimulus based on the preferred stimulus of the fastest responding neuron exhibits dramatically deteriorating accuracy as the number of alternatives grows larger [47]. Thus, the weak ability of the WTA model to achieve high accuracy, even when reading from large populations of neurons, and its difficulty addressing numerous alternatives raises doubts as to its ability to account for observed behavior in pop out tasks.

One potential alternative consists of a generalized population-based WTA algorithm that estimates the target location from the population with the strongest response rather than from the single cell with the strongest response. However, it remains unclear whether this type of mechanism can yield the high accuracy reported in pop out tasks.

Thus, on the one hand evidence supporting the WTA hypothesis has been presented [19,48–51], while on the other hand the accuracy of a WTA readout in a pop out task remains unclear. Here we conducted a modeling study to investigate the theoretical accuracy of both the single-best-cell and the generalized WTA in a pop out task. This theoretical accuracy can then be compared with the empirically estimated behavioral accuracy. Failure to account for the high empirical accuracy would thus signal a gap in our understanding. Since WTA accuracy may depend on a variety of parameters, we analyzed the ways in which the number of distractors, population size, contextual modulation strength, heterogeneity of the neuronal population and noise correlations affected the ability of both WTA and generalized WTA to identify the

deviant stimulus. Although fair estimates on single cell and pairwise statistics can be obtained empirically, the population size used for the decision can vary across several orders of magnitude. Consequently, special effort is allocated to understanding the scaling of WTA accuracy with population size.

## Results

This section is organized as follows. First, we present the basic toy model used here to generate a stochastic neural response to a pop out stimulus and analyze the accuracy of the single-best-cell WTA algorithm. Next, we report how we applied the WTA algorithm to an electrophysiological dataset of contextually modulated neurons. This is followed by our analysis of the effects of neuronal heterogeneity on single-best-cell WTA performance. We demonstrate that the readout accuracy of WTA is low in the presence of heterogeneity. We then turn to the generalized WTA and show that its accuracy is high when reading from large heterogeneous populations but that its accuracy is undermined by noise correlations.

### Model setup: single-best-cell WTA

We consider a model of $M + 1$ populations, or columns, of $N$ contextually modulated neurons each, responding to a pop out stimulus, Fig 1F.

**Visual stimulus:** The pop out stimulus consisted of $M + 1$ objects. One deviant object was designated as the target ($t$) and the others as $M$ distractors ($d$).

**Neural responses:** We modelled the statistical distribution of the neural responses of $M + 1$ populations (or columns) of $N$ neurons each. Each population $j \in 1, \ldots M + 1$ of neurons responded to a single object, either the target or a distractor. Unless otherwise stated (see below) we assumed that given a stimulus, the firing of different neurons was independent. Thus, given a stimulus, $\left\{ s_j \vert \, s_j \in t/d \right\}_{j=1}^{M+1}$, the joint probability (density) of the neural responses is given by:

$$\Pr\left( \left\{ x_i^j \right\} \vert \left\{ s_j \right\} \right) = \prod_{i=1}^{N} \prod_{j=1}^{M} f_{i,\, j}\left( x_{i,j} \vert s_j \right)$$

(1)

where, $x_{i,j}$ denotes the response of neuron $i$ in population $j$ during a single trial, such as the number of elicited spikes. In this work we do not incorporate finer details of the temporal structure of the response; hence, we model only the rate of firing. The function $f_{i,\, j}\left( x_{i,j} \vert s_j \right)$ is the marginal probability (density) of the response of neuron $i$ in population $j$, conditioned on the stimulus object in its receptive field, $s_j$. The function $F_{i,j}\left( x_{i,j} \vert s_j \right)$ is the cumulative distribution function, $F_{i,j}\left( x_{i,j} \vert s_j \right) = \int dx\, f_{i,j}\left( x_{i,j} \vert s_j \right)$.

We explored three types of distributions: Poisson, exponential and Gaussian distributions. Poisson and exponential distributions are typically considered a good approximation of neuronal response variability (see, e.g., [52–54]). The Gaussian distribution is useful for studying the effects of neuronal noise correlations. Unless stated otherwise, we take the variance of the Gaussian distribution to be equal to the mean.

We characterize the functions $f_{i,\, j}\left( x_{i,j} \vert s_j \right)$, $F_{i,j}\left( x_{i,j} \vert s_j \right)$ through two key metrics: the conditional mean response to the target object, denoted as $r_{i,j} = E\left[ x_{i,j} \vert s_j = t \right]$, and the contextual modulation strength, $q_{i,j}$, defined as the ratio of the mean responses to the target over the distractor: $q_{i,j} = \frac{E\left[ x_{i,j} \vert s_j = t \right]}{E\left[ x_{i,j} \vert s_j = d \right]}$. This modulation strength, $q_{i,j}$, quantifies the relative change in a neuron's response across different contexts and is an important parameter that serves as a proxy for information in a single neuron response (see Methods). The mutual information between the neuronal response and the stimulus is depicted, Fig 1H, as a function of the contextual modulation strength, $q$. In our stochastic model we selected $q$ based on electrophysiological data from the archerfish [21] (see Methods).

Many studies have investigated the origin of contextual modulation. One prominent theory posits that contextual modulation results from divisive normalization [55–57]. Here, our aim was not to determine the mechanism that generates contextual modulation. Rather, our interest lies in the accuracy at which WTA can extract information on the

stimulus so that our model simply implements contextual modulation extracted from this electrophysiological dataset (see Methods).

In this dataset, the mean contextual modulation strength value (over many neurons) was $\sim 1.44$, with typical contextual modulation strengths ranging from lower values of $\sim 1.1$ to $\sim 1.7$. Extreme or best typical contextually modulated neurons reach$\sim 2$ (Fig 1G) [21,34].

***The single-best-cell WTA readout (WTA):*** The task of the readout mechanism was to identify the deviant (target) stimulus based on the neural responses. Below, we analyze the accuracy of the single-best-cell WTA, which we term WTA for brevity. The winner is the neuron with the highest activity $x_{i,j}$ in the entire system of $j \in 1, \dots M+1$ populations of $i \in \{1, \dots N\}$ neurons each. The readout of the WTA is the stimulus, $s_j$, at the receptive field of the winning neuron. In case of a tie, the winner is selected randomly with equal probability from all the neurons with the strongest response.

## WTA performance in homogeneous populations

In the homogeneous case, the marginal response distributions (conditioned on the stimulus), $f_{i,j}(x|s_j = \alpha)$ [where $\alpha = t/d$], are identical for all neurons. Thus, $r_{i,j} = r_t$ and $q_{i,j} = q$, where the mean response to the target stimulus, $r_t$, and the strength of the contextual modulation, $q$, are the same for all neurons. For convenience we shall denote the mean response in a distractor population by $r_d = \frac{r_t}{q}$. Denoting $f_\alpha(x) = f_{i,j}(x|s_j = \alpha)$ [where $\alpha = t/d$], the accuracy of the WTA is given by:

$$P_c = N \int dx f_t(x) F_t(x)^{N-1} F_d(x)^{MN}$$

(2)

Where $F_\alpha(x)$ [$\alpha = t/d$] is the cumulative distribution function, $F_\alpha(x) = \int dx f_\alpha(x)$.

***Exponential distribution.*** In the case of an exponential response distribution, the analytical results for the dependence of the success rate, $P_c$, on the number of distractors, $M$, and the modulation strength, $q$, can be obtained in certain interesting limits.

***Exponential distribution N= 1.*** In the case of $N = 1$; that is, one neuron in each column, the accuracy of the WTA decision is given by (see Methods):

$$P_c = \frac{\Gamma(M+1)\Gamma\left(\frac{1}{q}\right)}{q\Gamma\left(M+1+\frac{1}{q}\right)}$$

(3)

In the limit of $q \to 1$, the readout accuracy approaches chance value, $P_c \to \frac{\Gamma(M+1)}{\Gamma(M+2)} = \frac{1}{M+1}$. For large $q$, and finite $M$ the accuracy converges to 1 algebraically in $q$, $P_c \approx \frac{q+\gamma-1}{q+\left(\ln(M+1)-\frac{1}{2(M+1)}\right)}$ (see Methods). In the limit of a large number of populations $M \gg 1$, the readout accuracy converges to zero as $P_c \approx \frac{\Gamma\left(1+\frac{1}{q}\right)}{q}\left(M+\frac{1+\frac{1}{q}}{2}\right)^{-\frac{1}{q}}$ (see Methods).

***Exponential distribution large N.*** In the limit of large $N$ and finite $M$, one obtains (see Methods):

$$P_c \propto 1 - qM\Gamma(q)N^{1-q}$$

(4)

Thus, the probability of success of the WTA converges to 1 algebraically fast in $N$ (Fig 2A). Nevertheless, the success rate decreases as the number of distractors, $M$, grows larger (Fig 2B). The only way to achieve independence with respect to the number of distractors is via a plateau effect; namely, when the performance approaches the maximal success rate of one, changes due to the number of distractors are small.

Specifically, denoting a tolerable level of error by $P_{critical}$, the plateau effect can be achieved for $N > N_{critical} = \left(\frac{M\Gamma(2+q)}{P_{critical}}\right)^{1/q}$ (Fig 2C). To account for empirical findings, one can select values of behavioral experiments in the range of $P_{critical} \sim 0.95 - 1$ (compare with Fig 1E), $M \sim 2-25$ [23,24] and values for $q$ from electrophysiology,

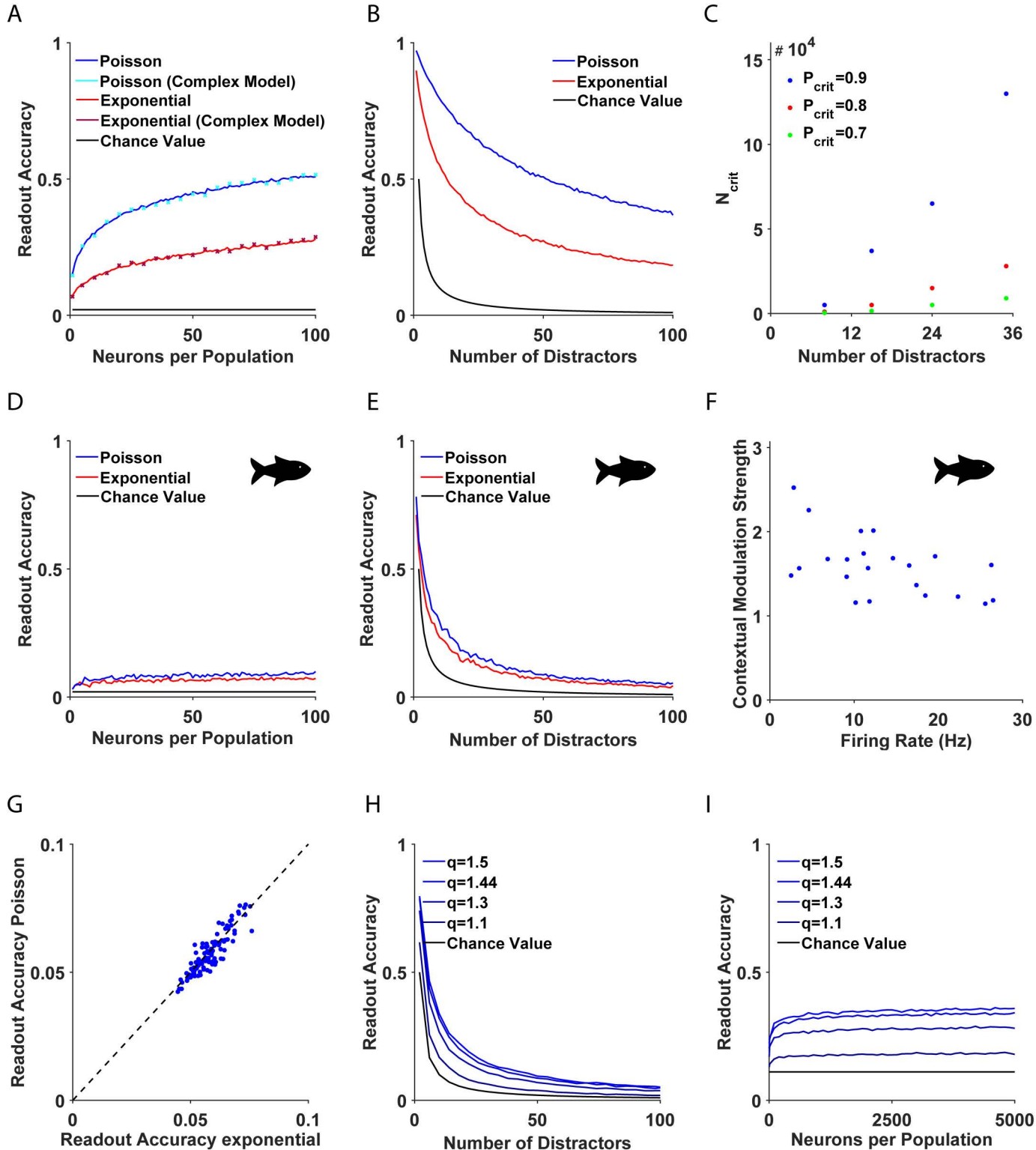

**Fig 2. Accuracy of the single-best-cell WTA.** (**A-B, D-E**) The accuracy of the single-best-cell WTA in homogeneous (**A-B**) and heterogeneous (**D-E**) systems is shown as a function of (**A, D**) the number of neurons, $N$, and (**B, E**) the number of distractors, $M$. The blue and red traces depict Poisson and exponential neuronal response distributions, respectively. Chance value is depicted in black. For a comprehensive list of all parameter values see Table

1 – Parameters for the numerical simulations. The scatter plot (cyan and magenta ($x$) markers) illustrates the readout accuracy of the WTA mechanism applied to natural images using the model proposed by Itti et al. [6], along with our framework (see Methods). (C) The number of neurons, $N_{critical}$, required to reach a given accuracy threshold level, $P_{critical}$, is shown as a function of the number of distractors, $M$. The different accuracy threshold levels are depicted by color. (F) Scatter plot depicting the response to a pop out stimulus and the contextual modulation strength of $22$ contextually modulated neurons in the optic tectum of the archerfish, where the correlation between the two parameters was $\rho = -0.53$, $p < 0.05$. Data courtesy of the Segev lab. The methodology and experimental procedure used to harvest these data are detailed in [21]. (G) WTA accuracy for a Poisson population is shown as a function of its accuracy for an exponential population; for the same realization of neuronal heterogeneity, the correlation was $\rho = 0.89, p < 0.05$. The identity mapping is presented (dashed blue line) for comparison. (H-I) WTA accuracy in artificial heterogeneous systems is shown as a function of (H) the number of distractors, $M$, and (I) the number of neurons, $N$, for different contextual modulation strengths, depicted by color. Chance value is depicted in black. The mean firing rates, $r_t$, and mean contextual modulation strength, $q$, used in A-E, and G are the same and were taken from the data, F.

$q \in [1.1, 1.5]$ [21] (see Methods). Using these parameters, we found that $N_{critical} \approx 10^4$ cells were needed to support the observed accuracy in the exponential case.

***Poisson response distribution.*** Qualitatively similar results were obtained for Poisson populations, as shown in Fig 1A and 1B (blue traces). In this case the WTA accuracy decayed to zero as the number of distractors, $M$, grew larger. In addition, its accuracy increased with the number of neurons per population, $N$, and a plateau was reached for $N > N_{critical} \approx 10^4$.

## WTA performance in the data-driven model

Next, we analyzed the WTA algorithm with parameters taken from electrophysiological recordings of contextually modulated neurons. The dataset consisted of $N_{data} = 22$ contextually modulated neurons from the optic tectum of archerfish responding to pop out and to uniform stimuli (Fig 2F) [21]. Each cell, $i \in 1, \ldots N_{data}$, in the dataset was characterized by a pair of values: its mean response to a pop out stimulus, $r_i$, (i.e., when the target was within its receptive field) and its contextual modulation strength, $q_i$.

We first generated a realization of an individual (see Methods). To do so, we chose randomly in an independent manner, with equal probabilities and with repetitions $N(M+1)$ neurons out of a pool of $N_{data}$ neurons, $\{(r_i, q_i)\}_{i=1}^{N_{data}}$. Each choice of $N(M+1)$ neurons represent an individual.

Note that there are two types of randomness in our model. The first is the trial-to-trial fluctuations that result from the stochastic neural response. The other is the frozen or 'quenched' disorder that describes fluctuations between different individuals.

The accuracy of the WTA is depicted as a function of the number of neurons in each column, $N$, and the number of distractors, $M$, in Fig 2D and 2E, respectively. The accuracy of the WTA was estimated numerically by averaging over the trial-to-trial fluctuations for each individual and then by averaging over $50$ realizations of different individuals.

Surprisingly, the accuracy of the WTA algorithm on this data-driven model was considerably lower than in the homogenous case (compare Fig 2A and 2B with 2D and 2E), even though both had the same average firing rate and contextual modulation strength. Furthermore, the rate at which WTA accumulated information from the large data-driven populations was also drastically reduced. The key difference can be attributed to the fact that actual neuronal populations are inherently heterogeneous, Fig 2F.

Interestingly, in the data-driven WTA model, the effect of the neuronal response distribution (e.g., Poisson or exponential) on its accuracy declined sharply (red and blue traces in Fig 2D and 2E). Fig 2G depicts WTA accuracy for Poisson and exponential response distributions for the same realization of individuals. As shown in Fig 2G, even though the WTA accuracy was slightly higher with Poisson statistics, in general the accuracy for the Poisson and exponential distributions was highly correlated ($\rho = 0.89$ , $p < 0.05$).

## The source of the failure of single-best-cell WTA in the data-driven model

Analysis of larger populations requires finding a solution to the finite size of the electrophysiological dataset. To overcome this issue, we generated artificial populations to study WTA performance for large heterogeneous systems that mimicked

the essential features of the electrophysiological dataset. The parameters, $r_{i,j}, q_{i,j}$, were drawn in an iid manner. Once $r_{i,j}$ and $q_{i,j}$ were drawn, they characterized the response of neuron $i$ in population $j$ and did not fluctuate from trial to trial. The set of parameters $r_{i,j}, q_{i,j}$ characterized an individual. Different individuals were characterized by a different realization of the parameters $\{r_{i,j}, q_{i,j}\}$. Specifically, we modelled $r_{i,j}$ following a log-normal distribution with mean and variance $r_t$ [58–60]. The contextual modulation strengths were drawn such that $\{q_{i,j} - 1\}$ were exponential random variables with mean $q - 1$.

We found that the accuracy of the WTA increased monotonically for both the mean strength of contextual modulation and the number of neurons, $N$. Nevertheless, even for populations of $N = 5000$ neurons, the performance of the WTA was very poor (Fig 2H and 2I). For example, the readout accuracy was $P_c \approx 0.35$ for $q = 1.44$ and $M = 8$ distractors, compared to the chance level of $\approx 0.11$.

Thus overall, even for large populations, accuracy was considerably lower than the behavioral data suggest. In the example above, the WTA failed more than $50\%$ of the trials whereas the behavioral data were well above the $95\%$ success rate.

To better understand the source of the poor performance of the WTA algorithm in data-driven heterogeneous populations, we briefly detour to examine what makes some individuals better than others. In the data-driven model, WTA accuracy was a random variable that depended on the specific realization of neuronal heterogeneity. Fig 3A and 3B show the confusion matrix for the WTA algorithm for two individuals in a pop out task with $M = 8$ distractors for two extreme cases. The confusion matrix element, $C_{i,j}$, is the probability of deciding the target location is at $i$, given that the target was at $j$. The diagonal, $\{C_{i,i}\}_{i=1}^{M+1}$, depicts the probability of correct identification (the hit rate) for different target locations, $i \in 1, \ldots M + 1$. For the first individual, Fig 3A, the hit rate varied from $0.2$ to $0.6$ depending on the target location. In the second example, Fig 3B, the hit rate was in the range $0.2$ to $0.35$. Thus, there was a variability in the performance across locations and across individuals. Furthermore, we found that the hit rate of target $i$, $C_{ii}$, was correlated ($\rho = 0.76$, $p < 0.05$) with the probability of erroneously estimating $j$ to be the target, $\frac{1}{M} \sum_{j=1, j \neq i}^{M+1} C_{ij}$, see Fig 3C.

What makes certain populations better than others at identifying targets? To shed light on this question, we examined which single neuron was responsible for the decision. In the WTA algorithm, a correct identification is said to occur when a single neuron in the target population is the winner. A high hit rate is expected when the decision is dominated by the activity of the more informative neurons; i.e., neurons that are characterized by high $q$ values. We defined the 'participation rate' of a neuron as the probability that the neuron was the 'winner', given a correct decision. In the simplest case of a homogeneous population, every neuron has the same probability of being the winner, and the participation rate is expected to be $\frac{1}{N}$.

In non-homogenous cases, the participation rate is highly non-uniform. For example, Fig 3D depicts the distribution of participation rates in a heterogeneous population of $N = 100$. Here, 17 neurons (17% of the population) were responsible for $\sim 50\%$ of the decisions (Fig 3E). In another example (Fig 3F) with $N = 1000$ neurons, fewer than $70$ neurons that made up 7% of the neural population were responsible for $50\%$ of the decisions, Fig 3G. Thus, a small fraction of the population was responsible for most of the decisions, and this fraction decreased as $N$ grew larger, Fig 3H. For example, for $N = 10000$ neurons, barely 1% of the population made $50\%$ of the decisions.

What characterizes neurons with a high participation rate? The analysis showed that the participation rate was not correlated with the contextual modulation strength, $q$, which is a proxy for the information content of single neurons (Fig 3I, ($\rho = 0.05$, $p = 0.67$)). Instead, neurons with the highest participation rate were the neurons with the highest mean firing rates (Fig 3J ($\rho = 0.89$, $p < 0.05$)).

However, in the electrophysiological dataset, these two characteristics of the neuronal response tended to be either uncorrelated or negatively correlated, as was shown for the archerfish (Fig 2F ($\rho = -0.53$, $p < 0.05$)). Hence, the poor performance of the WTA in heterogeneous systems can be attributed to the fact that the WTA algorithm estimates the target based on the most active neurons, which are roughly uncorrelated with the most informative ones.

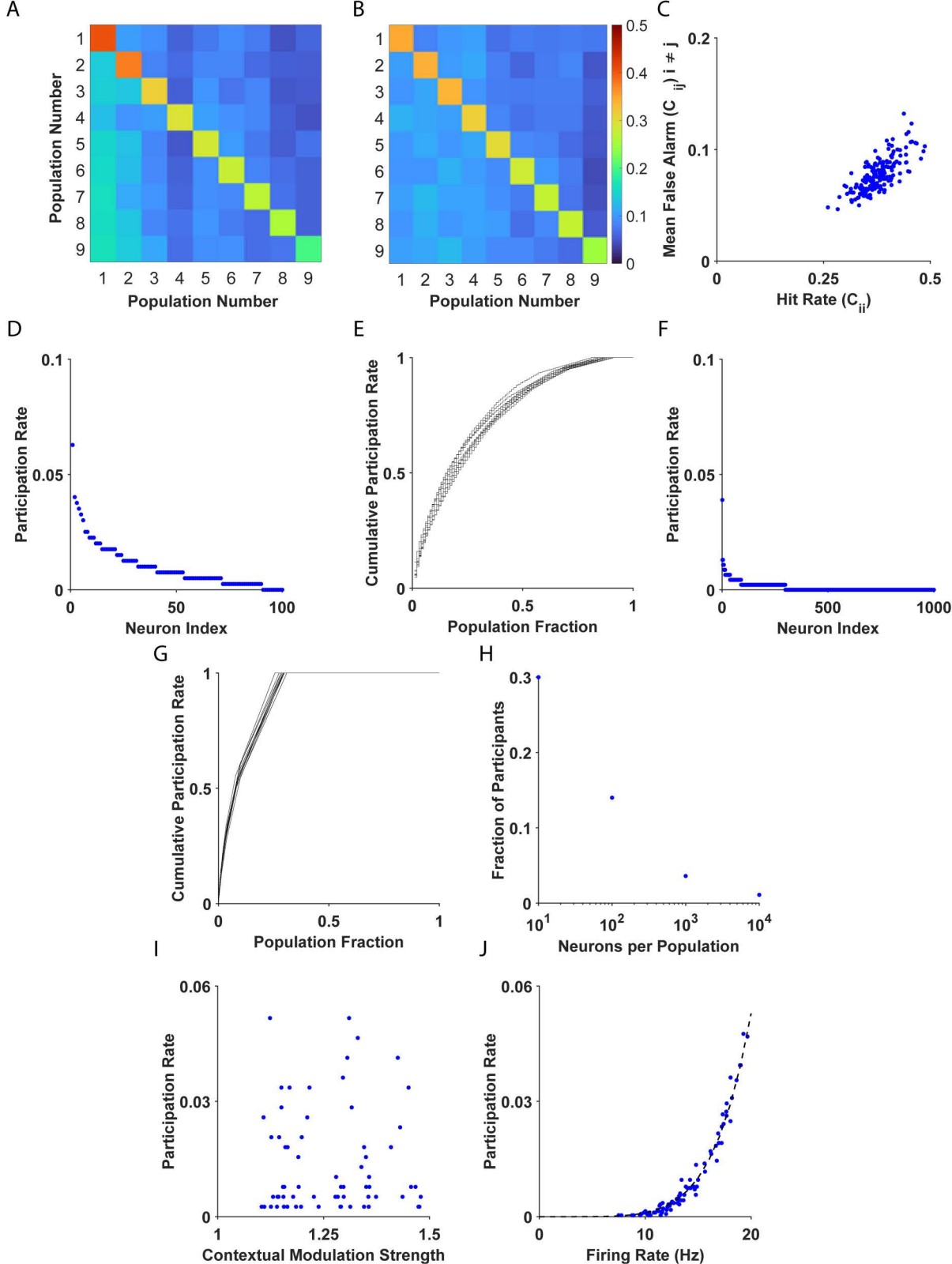

**Fig 3. Investigation of WTA errors. (A-B)** Two examples of confusion matrices presenting the performance of the WTA algorithm in two example heterogeneous systems. **(C)** The mean false alarm is shown as a function of the hit rate for different realizations of system heterogeneity; correlation

$\rho = 0.76$, $p < 0.05$. **(D)** Participation rate of different neurons is shown for one example of a single individual with $N = 100$ neurons per population. **(E)** The cumulative sum of the participation rate of the $n$ neurons in (D) with the highest participation rate, as a function of the fraction of neurons from the entire population, $n/N$. **(F)** As in D with $N = 1000$ neurons per population **(G)** As in E, for the neurons in **F. (H)** The fraction of neurons required to reach a cumulative participation rate of $50\%$ is shown as a function of the number of neurons per population, $N$. **(I)** Participation rate of different neurons is shown as a function of their contextual modulation strength, $\rho = 0.05$, $p = 0.67$. **(J)** Participation rate of different neurons is shown as a function of their firing rate. The dashed black line depicts the exponential fit of the form $f(x) = ax^b$, with $a = 1.77 \cdot 10^{-9}$, $b = 5.74$ and $R^2 = 0.98$.

### The poor performance of WTA is maintained in a more complex model

To verify that the results presented so far did not depend on the abstract system we analyzed, we next examined the accuracy of WTA when saliency was computed directly from an image (see Methods). For this purpose, we used the widely accepted model by Itti, Koch, and Niebur (2002) [6] of saliency-based visual attention. We used the code provided by the authors available at http://ilab.usc.edu/bu/, and applied biologically plausible parameters for the strength of contextual modulations to obtain the resulting variability of neuronal response. We found that the results were quantitatively similar to those obtained in the abstract system, (compare the x's and solid lines in Fig 2A). Thus, our central conclusion also holds for a more elaborate model.

### The generalized WTA model

One natural way to try to remedy the single-best-cell WTA algorithm is to consider competition between populations rather than between single neurons. In this generalized WTA competition, the stimulus is estimated by the receptive field of the most active population. The activity of population $j$ in response to stimuli $s_j = \alpha$ such that $\alpha = (t/d)$, is given by $Y_j = \frac{1}{N}\sum_{i=1}^{N} x_{i,j}$. Assuming the responses of different neurons are statistically independent given the stimulus, in the limit of large $N$, we can approximate the population activities by independent Gaussian random variables with $Y_j \sim G\left(R_j, \sqrt{\frac{R_j}{N}}\right)$, where $R_j = \frac{1}{N}\sum_{i=1}^{N} r_{i,j}$ for a target population and $R_j = \frac{1}{N}\sum_{i=1}^{N} \frac{r_{i,j}}{q_{i,j}}$ for distractor population. The variance of the single neuron response was taken to be equal to its mean [53,54]. A dynamical mechanism that realizes the generalized WTA was suggested in [61,62].

Fig 4A presents the results of the readout accuracy for the generalized WTA algorithm for heterogeneous populations. It shows that the generalized WTA accumulated information from large populations at a much faster rate than the WTA (compare Fig 4A to Fig 2D). This algorithm achieved an accuracy of $\sim 95\%$ using only a few hundred neurons per population for fixed $M$.

For a sufficiently large number of distractors, $M$, we can use the central limit for extreme values of a Gaussian distribution to approximate the cumulative distribution of the maximal value of activity of the $M$ distractor populations by a Gumble distribution, yielding:

$$P_c \approx \int_{-\infty}^{\infty} \sqrt{\frac{N}{2\pi R_t}} e^{\frac{-N(y-R_t)^2}{2R_t}} \; e^{-e^{-a(y-b)}} \; dy$$

(5)

with, $a = \sqrt{\frac{2qN\log(M)}{R_t}}$, and $b = \frac{R_t}{q} + \left(\sqrt{2\log(M)} - \frac{(\log(\log(M)) + \log(4\pi))}{2\sqrt{2\log(M)}}\right) \cdot \sqrt{\frac{R_t}{qN}}$

Where $R_t$ is the quenched average of the mean firing rate of the target population, and $q$ is the quenched average of the contextual modulation strength. The integrand of equation (5) consists of a Gaussian probability density multiplied by a Gumble distribution. For large $M$, we approximate the Gumble distribution by a Heaviside function centered around $b$, yielding:

$$P_c \approx \int_{b}^{\infty} \sqrt{\frac{N}{2\pi R_t}} e^{\frac{-N(y-R_t)^2}{2R_t}} \; dy = \frac{1}{2}\left(1 - erf\left(\sqrt{\frac{N}{2R_t}}(b - R_t)\right)\right)$$

(6)

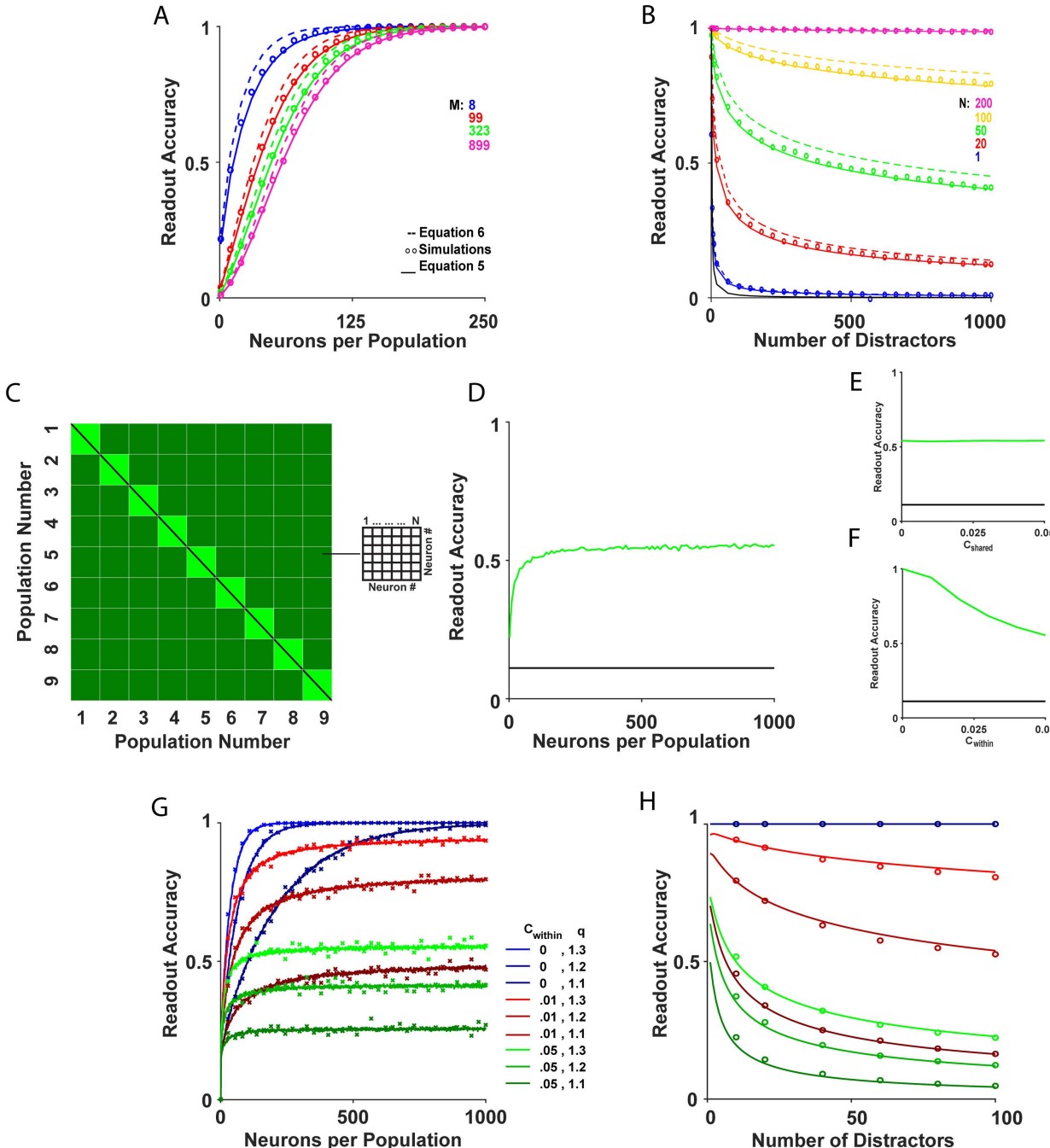

**Fig 4. Accuracy of the Generalized WTA. (A)** The accuracy of the generalized WTA is presented as a function of $N$. The different colors depict different numbers of distractors, $M$. **(B)** The accuracy is presented versus $M$. The different colors depict different values of $N$. The open circles, solid lines and dashed lines depict the accuracy as estimated by simulations, equation (5), and equation (6), respectively. Chance value is depicted in black. **(C)** A visual representation of the pairwise correlation matrix as defined in equation (9). This matrix captures the correlations among neurons within the system and has dimensions of $(N(M + 1))^2$. The diagonal elements (black) represent the variance of individual neurons. The central squares along the diagonal (light green) indicate the correlation coefficient $c_1$, whereas the remaining off-diagonal elements (dark green) indicate the correlation coefficient $c_2$. **(D-F)** Readout accuracy is presented in the cases of **(D)** both $c_{shared} \geq 0$ and $c_{within} \geq 0$ correlations, **(E)** $c_{within} = 0.05$, $c_{shared} \geq 0$, **(F)** $c_{within} \geq 0$, $c_{shared} = 0$. **(G)** Accuracy as a function of the number of neurons per population, $N$. The lines depict the numerical estimation of the accuracy in the Gaussian model. The x's depict the accuracy in the complex model. **(H)** Accuracy as a function of the number of distractors, $M$, for large $N = 10^4$. The solid lines show the analytical result of equation (5). The open circles depict the numerical estimation of the accuracy. The colors in G-H represent different correlation levels, $c_{within}$, and contextual modulation strengths, $q$.

where to a leading order in $M$, $b$ is given by:

$$b = \frac{R_t}{q} + \sqrt{\frac{2R_t \log(M)}{qN}} + O\left(\frac{\log(M)}{M}\right)$$

(7)

Note that for any fixed population size, $N$, the accuracy of the generalized WTA decreases to chance when the number of distractors increases. Nevertheless, the critical population size, $N_{critical}$, that reaches a certain level of accuracy, $P_{critical}$, depends logarithmically on the number of distractors, $M$. Thus, to observe a deterioration in the performance of the generalized WTA, the number of distractors needs to scale exponentially in $N$. We get:

$$N_{critical} \approx \left(\frac{\sqrt{2}\,erf^{-1}\left(2P_{critical} - 1\right) + \sqrt{\frac{2\log(M)}{q}}}{\sqrt{R_t}\left(1 - \frac{1}{q}\right)}\right)^2$$

(8)

As expected, for $q \to 1$ or for $P_{critical} \to 1$, one obtains $N_{critical} \to \infty$. A generalized WTA readout using several hundred neurons per population can achieve an accuracy of $98\%$ for $M \leq 35$ (Fig 4B).

Heterogeneity also affected generalized WTA performance. In heterogeneous systems, the trial-to-trial average activity of population $j$, $R_j$, fluctuated from one population to another due to the inherent neuronal heterogeneity with mean $\ll R_j \gg = \mathcal{O}(N^0)$ and variance $\ll R_j^2 \gg - \ll R_j \gg^2 = \mathcal{O}(\frac{1}{N})$, where the double angular brackets, $\ll \gg$, denote averaging with respect to the quenched disorder; i.e., the neuronal heterogeneity. Thus, averaging the neuronal responses across the population also reduced the effect of heterogeneity by a factor of $\frac{1}{\sqrt{N}}$. For example, compare the blue line in Fig 4A to Fig 2D.

## The source of the high accuracy of the generalized WTA model

This remarkable improvement in performance by the generalized WTA results from the fact that the magnitude of trial-to-trial fluctuations in the population responses was reduced by a factor of $\frac{1}{\sqrt{N}}$ due to spatial averaging. This hints that fluctuations that can generate a discrimination error are highly unlikely. However, this reasoning relies on the assumption that fluctuations in the responses of different neurons are uncorrelated. This fails to jibe with the literature reporting that noise correlations are widespread in the central nervous system [52,54,63–69], a topic that has elicited extensive theoretical analyses [70–74]. Note that some studies have only reported very weak correlations [75,76]. Other studies have found that noise correlations tend to be stronger between pairs of neurons with similar selectivity in the visual cortex [77–79], as expected in the column model we presented for the generalized case (Fig 1F). Thus, the effect of noise correlations on the accuracy of the generalized WTA must be considered.

## Noise correlations limit the accuracy of the generalized WTA

To study the effect of noise correlations on the accuracy of the generalized WTA, we modelled the response statistics of the system by a multivariate Gaussian distribution. Thus, given the stimulus, the neural responses $\{x_{i,j}\}$ follow a multivariate Gaussian distribution with means $r_{i,j} = E\left[x_{i,j}|s_j \in t/d\right]$, and covariance:

$$C_{ii'jj'} = \langle \delta x_{i,j} \delta x_{i'j'} \rangle = \sigma^2 \left(\delta_{ii'}\delta_{jj'} + c_1 \delta_{jj'}\left(1 - \delta_{ii'}\right) + c_2\left(1 - \delta_{jj'}\right)\right)$$

(9)

where $\sigma^2$ is the variance of the single cell response, $c_1$ is the correlation coefficient of the responses of different neurons from the same population and $c_2$ is the correlation coefficient between neurons from different populations, as

illustrated in Fig 4C. We assumed $c_1 \geq c_2$, since the correlation between pairs of neurons with overlapping receptive fields is typically larger than between pairs of neurons with non-overlapping receptive fields. This assumption is in line with electrophysiological findings reporting that neurons with closer tuning properties are characterized by higher correlations [65,66,77,79–81].

Fig 4D shows the accuracy of the generalized WTA as a function of the number of neurons per population, $N$, in a model with correlation, $c_1 \geq c_2 > 0$. As depicted in the Fig, the noise correlations imposed a limit on the asymptotic accuracy achievable by the generalized WTA (compare with Fig 4A).

To identify the source of this error, we partitioned the correlated part of the noise into two independent components (see also [63,82]): one where the noise was shared among all neurons across all populations, with a variance denoted $c_{shared}$ and the other where the noise within each population was not shared between populations, with variance $c_{within}$. Writing the response of neuron $i$ from population $j$ as the sum of these components yields: $x_{i,j} = r_{s_j} + \xi_{i,j} + \eta_j + \eta_0$, where $\xi_{i,j}$, $\eta_j$ and $\eta_0$ are independent Gaussian variables with zero means and variances: $Var[\xi_{i,j}] = \widetilde{\sigma}^2 = \sigma^2(1 - c_2)$, $Var[\eta_j] = \widetilde{\sigma}^2$, $c_{within} = \sigma^2(c_1 - c_2)$, and $Var[\eta_0] = \sigma^2$, $c_{shared} = \sigma^2 c_2$. Clearly, the shared component, $\eta_0$, cannot affect the generalized WTA decision. By subtracting the shared noise, $\eta_0$, from the responses of all neurons, the population activities become independent random Gaussian variables with means $r_t$ $\left(\frac{r_t}{q}\right)$ for target (distractor) population and variances $\widetilde{\sigma}^2\left(\frac{1}{N} + c_{within}\right)$, where we neglected the effect of neuronal heterogeneity, which vanishes for large $N$. Thus, in the limit of large $N$, the accuracy of the generalized WTA with $c_{within} > 0$, is equal to that of a system without correlations, $c_{within} = 0$, the same $\widetilde{\sigma}^2$, and $N_{eff} = \frac{1}{c_{within}}$. Note that $c_{within} = \frac{c_1 - c_2}{1 - c_2}$.

Fig 4E-F depict the asymptotic accuracy of the generalized WTA as a function of $c_{shared}$ and $c_{within}$, respectively. The analysis revealed that the accuracy of the generalized WTA was independent of $c_{shared}$, and reached an asymptotic value in the limit of large $N$ that was equal to the accuracy of an uncorrelated model with an effective population size of $N_{eff} = \frac{1}{c_{within}}$ (see Methods).

Fig 4G shows that models with the same positive within-population correlation value, $c_{within} > 0$ (compare different shades of green and red), approached a plateau roughly around the same $N \approx N_{eff}$. Similarly, the asymptotic accuracy of the generalized WTA and its dependance on the number of distractors, $M$, can be obtained from equation (5) by substituting $N$ for $N_{eff}$, Fig 4H.

Thus, within-population noise correlations appeared to limit the ability of the generalized WTA to accumulate information from large populations of neurons. Even correlations as weak as $c = 0.01$ caused the accuracy of the generalized WTA to saturate to $P_c \approx 90\%$, Fig 4G and 4H.

### The poor performance of generalized WTA persists in more complex models

To ensure that our findings were not limited to the abstract system we examined, we explored the accuracy of generalized WTA when saliency was directly computed from images (see Methods). We again utilized the established model of saliency-based visual attention developed by Itti et al. [6]. In our implementation we applied biologically plausible parameters for the strength of correlations, contextual modulations, and the heterogeneity of the neuronal response. As depicted in Fig 4G, the accuracy of the generalized WTA in the more complex model was quantitatively similar to that of the abstract system (compare the x's and lines).

### Discussion

Koch and Ullman's influential work laid the groundwork for the integration of the WTA mechanism as a pivotal tool in modeling selective visual attention [33]. Since then, these ideas have evolved into a fundamental cornerstone in the field. Building upon this initial foundation, their model was extended by introducing a more complex model that could effectively handle natural images [39, 83]. This constituted a significant stride forward in enhancing the WTA model's applicability.

Further progress was achieved by extending the model to address temporal visual stimuli, especially in the dynamic context of media such as movies [84]. These developments are characteristic of the ongoing refinement of selective visual attention models incorporating the foundational WTA mechanism. Thus, the WTA mechanism remains a natural choice for the computation of saliency [85–87] and decision-making [62,88].

We examined the accuracy at which a WTA-based algorithm could perform the pop out visual search task and compared it to the expected high behavioral accuracy. We considered two modes of the WTA algorithm. The first, the single-best-cell WTA, was based on WTA competition between contextually modulated cells. The second, generalized WTA, was based on competition between populations of contextually modulated cells.

We focused on the scaling of the accuracy of these WTA mechanisms with population size. Our rationale was that although the actual population size used in the decision is unknown, it is finite and limited. The population size in $V$1 can vary across different species by several orders of magnitude. Studies have shown for example that the estimated density of neurons per cubic millimeter [$mm^3$] is approximately $160,000$ in macaques [89], $40,000$ in cats [90], and $180,000$ in humans [91]. This makes the dependency of accuracy on population size critically important since plateau effects for accuracy can limit the feasibility of potential WTA mechanisms as possible explanations for pop out visual search.

We first analyzed the single-best-cell WTA by investigating the ability of this algorithm to correctly detect a single deviant object from a background of identical distractors. We found that the standard single-best-cell WTA accumulated information slowly as the neuronal population size increased. Note that the rate of improvement as a function of population size depends on the specific choice of the neuronal response distribution, since WTA is sensitive to the tail of the response distribution (see Methods). Thus, the accuracy in a Poisson population (blue) tends to be better than an exponential population (red), (Fig 2A and 2B).

When examining the inherent neuronal heterogeneity, this rate of integrating information was even slower. This is because the WTA decision is determined by the extreme values of the population response, which, as we showed in this work (Fig 3D–G), are dominated by a small percentage of cells from out of the entire population. Our data analysis of the electrophysiological findings in [21] indicated that the most active neurons that govern the single-best-cell WTA decision are not the most informative ones (Fig 2F). Overall, this makes the single-best-cell WTA an inappropriate model when aiming to account for the observed success rate in behavioral studies.

In an attempt to find a remedy for the failure of the single-best-cell WTA algorithm, we analyzed the generalized WTA algorithm. Here, the winning population is determined by a WTA competition between the average firing rates of each population. The analysis indicated that the generalized WTA success rate was greatly improved when there was competition between the mean (over the neural population in each column) responses. In addition, generalized WTA emerged as less sensitive to neuronal heterogeneity due to the spatial averaging.

However, the accuracy of the generalized WTA was limited by noise correlations within populations, $c_{within}$, but not by noise correlations shared by all neurons, $c_{shared}$. In the shared case, the correlations generated a collective mode of fluctuations in which the responses of all the neurons fluctuated together. Consequently, these collective fluctuations did not change the identity of the winner so that the shared correlations had little effect on accuracy.

By contrast, in the case of $c_{within}$, the correlations generated collective modes of fluctuations within the responses of the same population. Hence, these correlations affected the mean population response of separate populations differently and massively impacted the identity of the winner. As a result, the accuracy of the generalized WTA saturated to a size-independent limit, a highly constraining factor. Note that in systems without correlations (Fig 4A-B), high accuracy was achieved. However, the presence of correlations led to accuracy saturation (Fig 4G-H).

The contrasting impact of $c_{within}$ and $c_{shared}$ on WTA performance demonstrate that the structure of noise correlations, rather than their mere presence, critically determines whether they limit WTA decoding accuracy. Noise correlations give rise to large collective modes of fluctuations. If the signal used by the readout resides in the subspaces spanned by these collective modes, readout accuracy will be limited; hence the difference between the effect of $c_{within}$ and $c_{shared}$

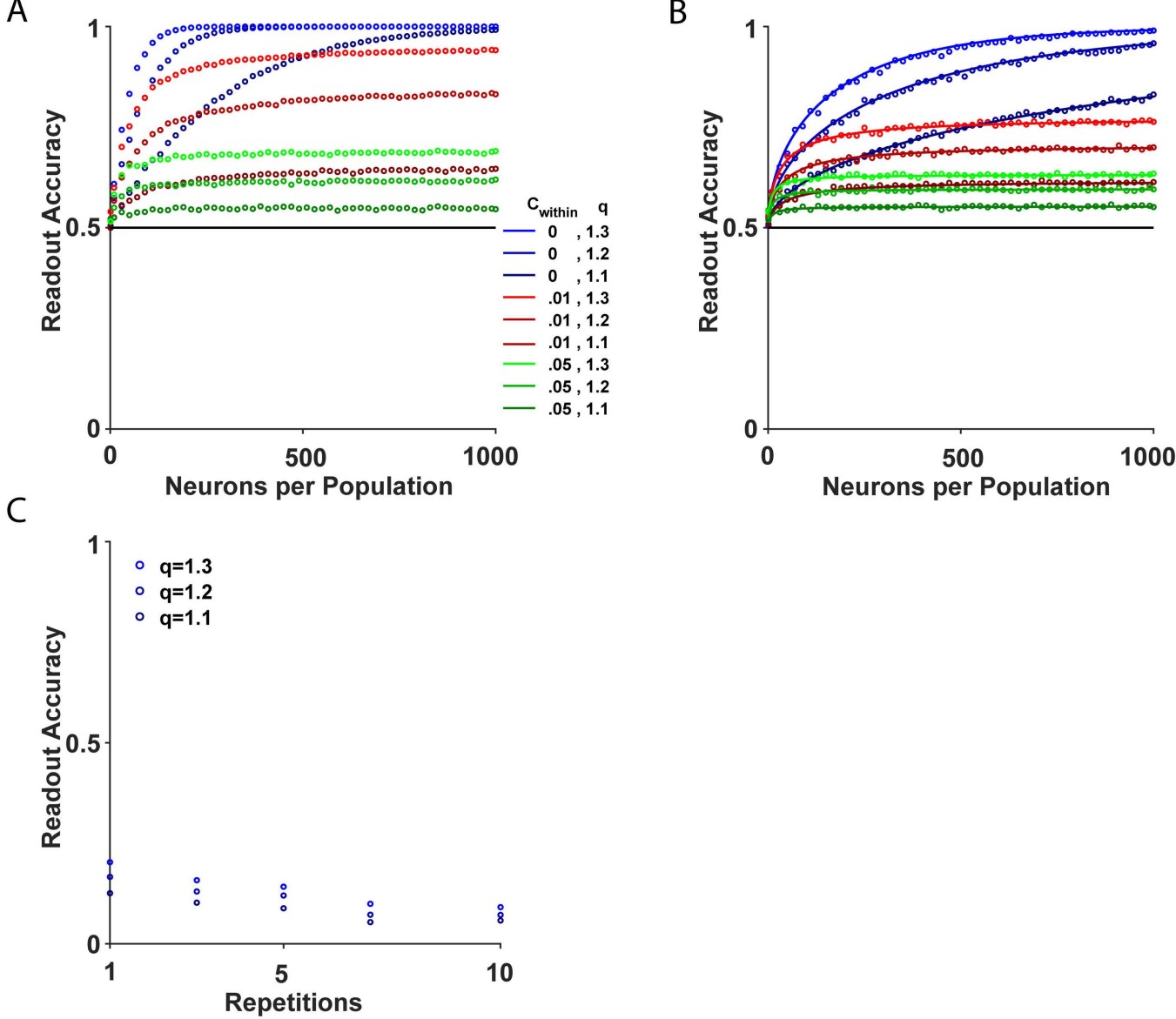

**Fig 5. Accuracy of WTA and generalized WTA in the present/absent task and repetition impact on WTA. (A-B)** Readout accuracy in the target present/target absent pop out task for **(A)** WTA, and **(B)** generalized WTA, is shown as a function of the number of neurons per population, $N$. Open circles and solid lines denote accuracy estimated via simulations and Equation (21), respectively. Different levels of within-population correlation, $c_{within}$, are indicated by different colors (blue, red, green), and contextual modulation strengths, $q$, are represented by shades of these colors. Chance value is depicted by a solid black line. **(C)** Readout accuracy of the WTA is plotted as a function of the number of repetitions, $K$ (see Methods), with different contextual modulation strengths, $q$, depicted by different shades of blue.

. Noise correlations can be shaped by shared inputs as well as by recurrent connections. Unlike in Itti & Koch [6], Li [30,31,92,93] used recurrent connectivity to obtain contextual modulation of neural responses. This suggests it would be of value to investigate the correlation structure generated by biologically plausible recurrent neural circuits, such as those in the primate V1 [94,95] or the fish optic tectum [96,97]. This critical question is beyond the scope of the current paper and will be addressed elsewhere.

**Present/absent task**

In this study, our primary focus up to this point has been on the accuracy of WTA in locating a pop out object. However, numerous studies have investigated the present/absent task, which examines whether a deviant object was present at all [5,24]. How well can the WTA identify the existence of a pop out stimulus? To this end, we presented two stimuli at random order. One, a pop out stimulus, consisted of one deviant object and $M$ identical distractors. The other consisted of $M + 1$ identical 'distractors'. The task of the readout was to identify in which interval the pop out stimulus was presented. Fig 5A-B depicts the accuracy of the WTA and generalized WTA on this task (see Methods for full details and analysis). As can be seen, the results for the present/absent task were qualitatively similar to the localization task.

**WTA accuracy is lower than psychophysical accuracy**

Observed success rates in saliency detection vary across different species. Humans, for instance, exhibit extremely high accuracy in saliency detection tasks, with success rates often reported above $96\%$ [9,23]. Non-human primates, such as macaques, also show impressive success rates, typically in the range $85 - 90\%$ [98]. Studies on archerfish show that these fish have success rates around $60\%$ [21,22]. Cats exhibit saliency detection success rates of approximately $90\%$ [99]. In contrast to these behavioral findings, here both the WTA and the generalized WTA models failed to achieve this high accuracy on the pop out task and fell considerably short of the performance seen in biological systems. This core result suggests that current understandings of the encoding and decoding of saliency should be revisited.

**Relationship of the findings to early vision and saliency maps**

The presence of saliency maps is a common theme in numerous studies of brain regions involved in bottom-up and top-down visual attention processing [6,31,38,39,100–105]. The key areas in bottom-up processes include the primary visual cortex ($V1$), the secondary visual cortex ($V2$), and the superior colliculus ($SC$). $V1$ contains neurons tuned to specific features with well-defined receptive fields, making them ideal for the spatial representation of external stimuli, which are often modelled by saliency maps [105,106]. $V2$ contains cells with larger receptive fields that exhibit tuning to more complex features compared to $V1$. Nevertheless, $V2$ also exhibits spatial coding of features, thus suggesting it may play an important role in saliency detection [107]. In addition, the $SC$ is thought to facilitate gaze shifts toward salient locations on the saliency map via the WTA competition, thus hinting at its involvement in pop out visual search [108–110]. Top-down visual processing may involve the $V4$ and the lateral intraparietal area ($LIP$), which are hypothesized to contain saliency maps [111–113]. This may be indicative of the possible role of saliency maps in higher cognitive modulation of attention. Specifically, the $LIP$ shows an increase in saliency map activity when a distractor is introduced [114], thus effectively shifting attention to prioritizing new targets, and possibly adjusting the focus in evolving visual scenes.

Thus, although many brain regions have been shown to be modulated by saliency, it remains unclear which specific area(s) of the brain compute saliency. Saliency maps, which are widely used to model visual attention across different brain regions [6,33,83,105,115], predominantly utilize rate-based models. However, our work reveals that these mechanisms may not fully account for the reported behavioral accuracy. If future electrophysiological studies provide new evidence that can challenge our basic assumptions on the statistics of the neural responses, this will greatly expand the theoretical possibilities for research in this field. For instance, noise correlations and contextual modulation strength may vary with the number of distractors. Alternatively, saliency may modulate spike timing, or other specific features of spiking network models that are not present in rate-based models, which may serve as additional source of information.

**Relationship to other visual search models**

**The Readout accuracy of the WTA mechanisms is also low in more complex models.** The Itti, Koch and Niebur model of saliency detection is extremely influential in the field [6]. One of the key features of their model is the utilization

of a WTA readout mechanism to determine salient locations in an image. In the current study, we introduced a framework to estimate the accuracy of WTA readout mechanisms. To validate our framework, we applied it to the saliency detection model proposed by Itti et al. [6], using their code. The results obtained in the complex model were consistent with our analytical findings. Specifically, the accuracy observed in both WTA mechanisms (single-best-cell WTA and generalized population-based WTA) failed to coincide with the high accuracy observed in behavioral experiments. These findings suggest that the limitations and discrepancies observed in the accuracy of the WTA mechanisms are not specific to any single implementation and point to broader challenges inherent to saliency detection methods that are rooted in biological constraints such as heterogeneity and noise correlations.

**Evidence for a WTA-like mechanism in the primary visual cortex.** An alternative theory by Li [30] to that of Itti and colleagues [6] proposes that saliency is computed directly within the primary visual cortex ($V1$), without requiring the integration of multiple maps. Supporting evidence includes zero-parameter predictions of human response times [19,48], showing that $V1$ activity alone can account for performance in pop out tasks, with additional validation from electrophysiological recordings in the monkey $V1$ [20], where reaction times were reliably predicted from neural activity. These important findings deepen the puzzle. To address this issue, it is important to stress that our conclusions as to the poor performance of the WTA are only valid under the constraints of our model's assumptions. These assumptions broadly fall into two categories: the specific implementation of the WTA algorithm and the statistical properties of the neuronal responses. Below, we discuss several possible solutions to this conundrum; namely, how a WTA-like algorithm can achieve higher accuracy in a pop out task.

**Using a better readout algorithm.** As early as the $1980$'s researchers demonstrated that relying on a single-stage parallel process in visual search (not necessarily pop out) can introduce errors, particularly as the set size increases. To address this issue, additional verification steps consecutive to the error-prone parallel process have been proposed [12,116–118]. Similar mechanisms are incorporated into many visual search models, such as Wolfe's Guided Search models ($2.0$ [7] and $6.0$ [87]). However, these additional steps come at the cost of increased computation time. How can the WTA algorithm be improved under the constraint of fast decisions?

We tested the readout accuracy of a repeated application of the WTA under the constraint of a fast decision. To this end, we partitioned the entire observation time of $T = 200 \, [ms]$ into $k$ intervals of equal duration. In each interval a winner was selected, where the winner of the repeated application was the location that won the most (see Methods). We found that for populations of Poisson neurons this approach failed to yield better accuracy (Fig 5C). Nevertheless, a more sophisticated integration of information could improve performance; for example, iterative strategies that refine candidate selection over multiple iterations, see [116]. Below, we discuss additional possibilities to increase WTA performance.

## Possible extensions of our work

Here we only considered a generalized WTA that utilized the mean neural response in each population. In this case, the response of each neuron had the same weight in the decision. However, due to neuronal heterogeneity, some neurons are more informative than others, such that the mean response could usefully be replaced with a weighted average. Previously it was shown that a readout algorithm that takes this diversity into account can overcome the limiting effect of noise correlations [67,70,119]. However, this approach requires some degree of fine-tuning of the weights. Investigation of this type of algorithm is beyond the scope of the current work and will be addressed in the future.

In addition, two central features were not analyzed in the current study: the temporal aspect of the decision and the dynamical system that implements the computation. The accuracy of a temporal generalization of the WTA has been studied in the framework of a two-alternative forced choice task [46,47]. It was shown that this generalization can yield high accuracy when required to discriminate between a small number of alternatives but fails when the number of alternatives is large. This generalization can be considered as a race-to-threshold decision mechanism [42] and can be implemented by using a simple reciprocal inhibition architecture [61,62]. The issue of the accuracy of a dynamical implementation

of a race-to-threshold WTA decision mechanism in a pop out task is beyond the scope of the current work and will be addressed elsewhere; but see [62]. However, a theoretical investigation of this kind requires empirical data on the contextual modulation of the temporal response, which is currently lacking. Consequently, even in studies that attempt to model neuronal activity with spiking frameworks, the underlying calculations revert to rate-based models [6,120–123].

### Level of abstraction

Selection of the proper abstraction level is an open issue in systems neuroscience. In our work we chose to investigate the performance of the WTA mechanism using a highly simplified model to facilitate mathematical analysis. It is crucial to realize that our simplifying assumptions create limitations. We investigated the WTA competition between populations of neurons within the same brain region. These populations were assumed to be statistically identical (i.e., same distribution of firing rates and contextual modulation strengths). Thus, we modeled neural populations that were selective to the same visual feature (orientation, shape, color etc.). However, in biological systems, the computation of saliency involves different brain regions and richer stimuli. Further, our model did not incorporate the intricate feed-forward, feed-back and recurrent connectivity characteristics of the central nervous system [30,32,124–128]. Instead, we modeled the statistics of the resultant neural responses. Finally, our study employed simplified non-spiking neurons. The choice of a rate model over a spiking network model offers a tractable framework for exploring neural computation but fails to capture the richness and complexity of biological neurons.

Given all the above, the inherent tradeoff between abstraction and biological realism emphasizes the need for caution in extrapolating our findings. Addressing these limitations is beyond the scope of the current study and will be addressed elsewhere. Nevertheless, this work highlights the essential features of neuronal response statistics that should be further investigated. These include the distribution of firing rates, contextual modulation strengths and correlations, the structure of the noise correlation and its dependence on the stimulus, as well as the temporal aspects of the neural dynamic response to the stimulus.

### Directions for future empirical research

Our work highlights a gap in current theorizing; namely, that WTA mechanisms fail to account for the high accuracy observed in pop out tasks despite strong empirical evidence supporting their role in saliency detection [19,48]. However, these findings only hold under our model's assumptions about the statistics of the neural response to a pop out stimulus. Would contextual modulation strength be higher at very short time intervals? Would the effect of noise-correlations decrease with the number of distractors? Could contextual modulation and noise correlations at shorter time intervals (the first $100$ [$ms$] following stimulus presentation) differ from those measured at longer intervals (see for example [129])? How is the temporal aspect of the neuronal responses modulated by the context? All these features are likely to have a significant effect on the accuracy of the WTA algorithm. We hope that our work will motivate further research in these directions, which in turn can shed light on both the algorithm and the brain regions where saliency is computed.

## Methods

### Quantifying information in single neuron response

Mutual information quantifies how much information can be obtained about the stimulus by observing the neural response. Formally, the mutual information between the neural response $x$ and the stimulus $s$ is given by:

$I(x,s) = \int_0^\infty dx \sum_s p(x,s) \log\left(\frac{p(x|s)}{p(x)}\right)$ where the stimulus can be either target or distractor $s \in \{t, d\}$ with probability $p$ and $1-p$. However, the calculation of the mutual information is cumbersome and in Fig 1H was estimated numerically with $p = \frac{1}{2}$.

To better understand how contextual modulation strength q affects information, it is convenient to study related measures such as the Kullback-Leibler divergence. The Kullback-Leibler divergence between the response distribution to

target versus to distractor stimulus, $D_{kl}$, can serve as a proxy for information content. For an exponential response distribution, we have $D_{kl} = \ln\left(\frac{1}{q}\right) + q - 1$, for Poisson $D_{kl} = r_t\left(1 - \frac{1}{q}\right) + \frac{1}{q}\ln\left(\frac{1}{q}\right)$, and for Gaussian $D_{kl} = \frac{r_t^2\left(1-\frac{1}{q}\right)^2}{2\sigma^2}$, where $r_t$ and $\sigma^2$ are the mean response and variance of the response of a given neuron to the target stimulus, correspondingly. Thus, the Kullback-Leibler divergence is zero when the neural response is independent of the stimulus, $D_{kl}(q=1) = 0$ and increases as the contextual modulation strength diverges from $1$.

**WTA accuracy for exponential distribution, in the case of $N = 1$**

For the simple case of one neuron per population, $N = 1$, the WTA accuracy is:

$$P_c = \int_0^\infty dx \frac{e^{-\frac{x}{r_t}}}{r_t}\left[1 - e^{-\frac{qx}{r_t}}\right]^M \tag{10}$$

Where, $M$ is the number of distractor populations, $r_t$ is, the mean response of the neuron to target stimulus and $q$ is the contextual modulation strength. By a change in variable $y \equiv e^{-\frac{qx}{r_t}}$, one obtains:

$$P_c = q^{-1}\int_0^1 dy[1-y]^M y^{\frac{1-q}{q}} = \frac{\Gamma(M+1)\Gamma\left(\frac{1}{q}\right)}{q\Gamma\left(M+1+\frac{1}{q}\right)} \tag{11}$$

In the limit of $q \to 1$, $P_c$ converges to the chance value, $P_c = \frac{\Gamma(M+1)}{\Gamma(M+2)} = \frac{1}{M+1}$. To study the limit of large $q$ for finite $M$, we used the first order expansion of the gamma function: $\Gamma\left(M+1+\frac{1}{q}\right) = \Gamma(M+1) + \Gamma(M+1)\psi^0(M+1) \cdot \frac{1}{q}$, where $\psi^0$ is the zero order of the polygamma function. Next, we expanded the polygamma function as $\psi^0(M+1) \approx \ln(M+1) - \frac{1}{2(M+1)}$. Using the above approximation yields $\frac{\Gamma\left(M+1+\frac{1}{q}\right)}{\Gamma(M+1)} \approx 1 + \left(\ln(M+1) - \frac{1}{2(M+1)}\right) \cdot \frac{1}{q}$. For large $q$ we can approximate $\Gamma\left(\frac{1}{q}\right) \approx q + \gamma - 1$, where $\gamma$ is the Euler-Mascheroni constant, obtaining:

$$P_c \approx \frac{q + \gamma - 1}{q + \left(\ln(M+1) - \frac{1}{2(M+1)}\right)} \tag{12}$$

Thus, for finite $M$, $P_c$ approaches $1$ algebraically in $q$.

For $M \gg 1$, using the asymptotic expansion of the gamma function one obtains $\frac{\Gamma(M+1)}{\Gamma\left(M+1+\frac{1}{q}\right)} \to \left(M + \frac{q+1}{2q}\right)^{-\frac{1}{q}}$, which yields:

$$P_c \approx \frac{1}{q}\Gamma\left(\frac{1}{q}\right)\left(M + \frac{q+1}{2q}\right)^{-\frac{1}{q}} \tag{13}$$

Thus, for any fixed $q > 1$, $P_c$ decays to chance algebraically in $M$.

**WTA accuracy for an exponential distribution, with large $N$**

Changing variables to $y \equiv 1 - F_t(x)$ in equation (2) yields in the homogeneous case:

$$P_c \approx N\int_0^1 dy \frac{e^{Ng(y)}}{1-y} \tag{14}$$

$$g(y) = log[1-y] + M\log\left[1 - y^q\right] \tag{15}$$

For large $N$, this integral is dominated by small $y$. Substituting $u = -g(y)$, and noting that for small $y$, $(i)\ u \approx y + My^q$ (for finite $M$ in the limit of large $N$), and $(ii)\ \frac{du}{dy} \approx \frac{1}{1-y}\left[1 + qMy^{q-1}\right]$, we obtain:

$$P_c = N \int_0^\infty du \frac{e^{-Nu}}{1 + qMu^{q-1}} \approx 1 - Mq\Gamma(q)N^{1-q}$$

(16)

where we used Watson's lemma to obtain the last result.

### WTA error for different noise types

Decision errors in the WTA algorithm result from large fluctuations in the neural responses. The likelihood of these fluctuations depends on the tail of the response distribution. The tail of the distribution behaves differently for exponential, $P(x) \sim e^{-xr}$, Poisson, $P(x) \sim e^{-x \cdot \ln(x)}$, and Gaussian, $P(x) \sim e^{\left(-\frac{(x-r)^2}{2\sigma^2}\right)}$ distributions (where, we denoted by $r, \sigma^2$, the mean and variance of the neurons response to the stimulus). Consequently, for a similar mean and variance, the WTA algorithm is expected to perform better for an exponential population of neurons than for Poisson, and for Poisson better than for Gaussian. Compare for example the red and blue traces in Fig 2A–B.

### WTA accuracy in the present/absent task

**The task.** We modeled a two-interval-two-alternative forced choice task. One interval was a pop out stimulus where the visual stimulus contained $M$ identical objects and one deviant object. In the other interval the stimulus was composed of $M + 1$ identical objects. The task of the readout was to infer the interval in which the pop out stimulus was presented.

**Statistical model of the neuronal response.** We modeled the responses of $M + 1$ populations or columns of $N$ neurons each. We assumed that, given the stimulus, the responses to the first and second intervals were independent. For simplicity we assumed that each population was homogeneous. Thus, the response, $x_{i,j}^l$, of neuron $i \in \{1, \ldots, N\}$ in population $j \in \{1, \ldots, M + 1\}$ to stimulus $s_j^l \in \{t, d\}$ during interval $l \in \{1, 2\}$, had a marginal response distribution, conditioned on the stimulus, $f\left(x_{i,j}^l | s_j^l\right)$. The conditional mean response to the pop out target object is given by $r_t = E\left[x_{i,j}^l | s_j = t\right]$, and the contextual modulation strength is the ratio $q = \frac{E\left[x_{i,j}^l | s_j = t\right]}{E\left[x_{i,j}^l | s_j = d\right]}$.

In this scenario, we can simply write: $f_\alpha(x) = f_{i,j}^l\left(x | s_j^l = \alpha\right), [\alpha = t/d]$.

**Definition of the readout.** WTA estimates the target interval by the interval with the neuron with the highest response. In the event of a tie, one of the tied neurons is selected uniformly at random (the probability of a tie is zero for continuous random variables, see below).

**Readout accuracy.** The probability of correctly identifying the interval with the pop out stimulus is given by:

$$P_c = N \int dx f_t(x) F_t(x)^{N-1} F_d(x)^{(2M+1)N} + MN \int dx f_d(x) F_t(x)^N F_d(x)^{(2M+1)N-1}$$

(17)

where $f_\alpha(x) = f_{i,j}^l\left(x | s_j^l = \alpha\right), [\alpha = t/d]$, and $F_\alpha(x)$ is the corresponding cumulative distribution.

**Exponential population.** Assuming that given the stimulus, the responses of different neurons are independent and that their marginal distribution follows exponential statistics, we can apply Watson's lemma (see also above, equation (16)) to approximate each integral by:

$$N \int dx f_t(x) F_t(x)^{N-1} F_d(x)^{(2M+1)N} \approx 1 - (2M+1)q\Gamma(q)N^{1-q}$$

(18)

and,

$$MN \int dx f_d(x) F_t(x)^N F_d(x)^{(2M+1)N-1} \approx M\, q\, \Gamma(q) N^{1-q} \tag{19}$$

Combining Equation (18) and (19) we obtain:

$$P_c \approx 1 - (3M+1)q\Gamma(q)N^{1-q} \tag{20}$$

**Generalized WTA accuracy in the present/absent task**

In the present/absent task the generalized WTA estimates the interval with the pop out stimulus by the interval with the largest mean activity, $Y^I = \frac{1}{N(M+1)} \sum_{j=1}^{M+1} \sum_{i=1}^{N} x_{i,j}^I$. Thus, the generalized WTA estimates the pop out was presented in interval $1$ if the decision variable, $q \equiv Y^1 - Y^2$, is positive and $2$ if negative. Thus, given the stimulus, the neural responses are assumed to follow a multivariate Gaussian distribution with noise correlations within each population $c_{within}$. Denoting $x_{i,j}^I = r_{s_j}^I + \xi_{i,j}^I + \eta_j^I$, where the superscript '$I$' denotes the interval, and $\xi_{i,j}^I$ and $\eta_j^I$ are independent Gaussian variables with zero means and variances: $Var\left[\xi_{i,j}^I\right] = \widetilde{\sigma}^2 = \sigma^2 (1 - c_2)$, $Var\left[\eta_j^I\right] = \widetilde{\sigma}^2 c_{within} = \sigma^2 (c_1 - c_2)$, where $c_1$ and $c_2$ are as defined in equation (9). Let $N \gg 1$, the $Y^1$ and $Y^2$ be independent random Gaussian variables with means $r_t$ and $\frac{r_t}{q}$ for target and distractor population respectively, and with variance $\widetilde{\sigma}^2 \left(\frac{1}{N} + c_{within}\right)$.

The readout accuracy of the generalized WTA can be calculated analytically, neglecting the quenched disorder, yielding:

$$P_c = \frac{1}{2} erfc \left( \frac{-r_t \left(1 - \frac{1}{q}\right)}{\sqrt{2(2M+2)\left(\frac{1}{N} + c_{within}\right)} \cdot \widetilde{\sigma}} \right) \tag{21}$$

In Fig 5B, we compare the accuracy of the simulations (depicted by circles) to the analytical results (solid line). Different colors correspond to varying correlation strengths, while different shades represent distinct values of the contextual modulation strength. Notably, even for correlations as low as $1\%$ (green), the accuracy remains low. Note that the accuracy of the generalized WTA with $c_{within} > 0$, is equal to that of a system without correlations, $c_{within} = 0$, the same $\widetilde{\sigma}^2$, and $N_{eff} = \frac{1}{c_{within}}$.

## Numerical methods

### Experimental data-driven system

The data used in this section was provided by the Segev lab. The methodology and experimental procedure used to harvest these data are detailed in [21]. The dataset contains $65$ neurons, out of which we selected $22$ neurons that showed clear contextual modulation. To this end we used the following criterion. We chose neurons whose estimated mean firing rate in the pop out condition minus twice the standard error of its mean were greater than the estimated mean firing rate in the uniform condition plus twice the standard error of this mean.

We characterized each of the $22$ neurons by two key parameters: its mean firing rate in response to a pop out stimulus and its contextual modulation strength. The mean firing rate of the neuronal population to pop out and uniform stimuli was $12.9\ Hz$ and $8.9\ Hz$, respectively, and the mean contextual modulation strength $q$ was $1.44$ (see Fig 2F). However, the firing rates of individual neurons were widely distributed around the population average, Fig 2F.

To generate a realization of an individual system consisting of $M + 1$ populations of $N$ neurons each, we drew $N(M + 1)$ neurons from the set of $22$ contextually modulated cells randomly with equal probabilities and with replacement. To generate a single trial of the neural response to a pop out stimulus, the deviant population was chosen first. Then, the spike count for each neuron was drawn independently from either a Poisson or an exponential distribution. A time window of $200$ [$ms$] [130] was used to translate the mean firing rate of each neuron to the mean spike count response to the stimulus.

## Generation of artificial systems with biological constraints

To generate a system of $M + 1$ populations with $N$ neurons each, we drew the firing rates in response to target stimulus, $r_{ij}$, and contextual modulation strengths, $q_{ij}$, for each of the $N(M + 1)$ neurons randomly in an independent manner. Specifically, the firing rates in response to a pop out stimulus were drawn from a log-normal distribution [53,58–60,131–133] with mean $\mu = 12.8$, and standard deviation $\sigma = 3.57$. The contextual modulation strengths were drawn from an exponential distribution, with mean $1.44$, as in the electrophysiological dataset. The accuracy of the WTA in a single realization of an individual (represented by a single system of $N(M + 1)$ neurons) was estimated by simulating the response over the course of $1000$ trials. In our analysis, we specifically tested three types of trial-to-trial response statistics: Poisson, exponential, and Gaussian.

## Artificial systems with correlations

To construct a system consisting of $M + 1$ populations, with each population comprising $N$ neurons, we independently sampled the firing rates in response to a target stimulus, denoted as $r_{ij}$, and the contextual modulation strengths, represented as $q_{ij}$, for each of the $N(M + 1)$ neurons. Our approach relied on the utilization of Gaussian random variables characterized by mean $r_{ij}$ and variance $r_{ij}$, alongside a covariance matrix $C$ as described in equation (9). To draw the values $r_{ij}$ we used a log-normal distribution [58,59], with a mean of $r_t = 12.8$ and a variance equal to the mean [53,54]. The variable $q$ followed an exponential distribution with a rate parameter of $1.3$, $1.2$, or $1.1$. We conducted $20$ realizations, with $1000$ trials each.

## Present/absent task

We simulated a two-interval-two-alternative forced choice present/absent task, in which one interval contained a pop out target, and the other did not. The task of the readout was to determine in which interval the pop out was present. We used one of two readout strategies: $(1)$ the WTA readout, which estimated the interval in terms of the one with the single most active neuron, $(2)$ the generalized WTA readout, which estimated the interval in terms of the interval with the highest mean neuronal activity across the entire population of $M + 1$ populations or columns. To simulate the neuronal responses to the stimuli we utilized the neuronal response statistics described above in the *Artificial systems with correlations* section. For each set of parameters $(N, q, c_{within})$, the readout accuracy was estimated by averaging over $20$ realizations of inherent neuronal heterogeneity and for each realization over $1000$ trials of the neuronal stochastic response.

## Time window for saliency computation

One parameter that cannot be well-estimated is the processing or computation time; i.e., the duration of information integration from the contextually modulated cells. Longer processing times make the single neuron response more informative, since both the mean and the variance of the spike count are expected to scale linearly with the processing time. As a result, the accuracy of both WTA and generalized WTA will also improve.

What is a reasonable estimate for processing time? One can bound processing time from above by the reaction time, which is on the order of $500$ [$ms$] and includes both the delay of the sensory response and the motor output. A tighter estimate of processing time was obtained by Stanford and Salinas in a two-alternative forced choice task [130]. Using a cleverly designed experiment, they estimated processing time to be on the order of a few tens of milliseconds (see also [134]). Here, we used a conservative estimate of $200$ [$ms$] for the processing time. Shorter durations only further decreased the accuracy of the WTA.

## Repeated application of the WTA

To investigate the repeated application of the WTA mechanism, we simulated a system consisting of $M+1$ populations, each containing $N$ neurons. Each neuron was characterized by two parameters: its mean response to a pop out stimulus, $r_{i,j}$, and its contextual modulation strength, $q_{i,j}$. The values of $r_{i,j}$ were independently drawn from a log-normal distribution with a mean of $12.8$ and a standard deviation of $3.5$, whereas the $q_{i,j}$ values were independently sampled from an exponential distribution with a mean of $1.3$, $1.2$, or $1.1$. The neural responses were assumed to follow homogeneous Poisson process statistics.

**Table 1. Parameters for numerical simulations.**

| Fig 2 panel | N | M | $r_t$ | q | Trials | Realizations |
|---|---|---|---|---|---|---|
| A | | 48 | 12.8 | 1.44 | 1000 | 20 |
| B | 100 | | 12.8 | 1.44 | 1000 | 20 |
| C | | | 12.8 | 1.44 | 1000 | 20 |
| D | | 48 | $\log(x) \sim N(12.8, 12.8)$ | $x \sim \exp(\lambda = 1.44)$ | 1000 | 50 |
| E | 100 | | $\log(x) \sim N(12.8, 12.8)$ | $x \sim \exp(\lambda = 1.44)$ | 1000 | 50 |
| G | 100 | 48 | $\log(x) \sim N(12.8, 12.8)$ | $x \sim \exp(\lambda = 1.44)$ | 10000 | 1 |
| H | | 8 | $\log(x) \sim N(12.8, 12.8)$ | | 1000 | 20 |
| I | 5000 | | $\log(x) \sim N(12.8, 12.8)$ | | 1000 | 20 |
| **Fig 3 panel** | | | | | | |
| A | 100 | 8 | $\log(x) \sim N(12.8, 12.8)$ | $x \sim \exp(\lambda = 1.44)$ | 1000 | 1 |
| B | 100 | 8 | $\log(x) \sim N(12.8, 12.8)$ | $x \sim \exp(\lambda = 1.44)$ | 1000 | 1 |
| C | 100 | 8 | $\log(x) \sim N(12.8, 12.8)$ | $x \sim \exp(\lambda = 1.44)$ | 1000 | 20 |
| D | 100 | 8 | $\log(x) \sim N(12.8, 12.8)$ | $x \sim \exp(\lambda = 1.44)$ | 1000 | 1 |
| E | 100 | 8 | $\log(x) \sim N(12.8, 12.8)$ | $x \sim \exp(\lambda = 1.44)$ | 1000 | 1 |
| F | 1000 | 8 | $\log(x) \sim N(12.8, 12.8)$ | $x \sim \exp(\lambda = 1.44)$ | 1000 | 1 |
| G | 1000 | 8 | $\log(x) \sim N(12.8, 12.8)$ | $x \sim \exp(\lambda = 1.44)$ | 1000 | 1 |
| H | | 8 | $\log(x) \sim N(12.8, 12.8)$ | $x \sim \exp(\lambda = 1.44)$ | 1000 | 20 |
| I | 1000 | 8 | $\log(x) \sim N(12.8, 12.8)$ | $x \sim \exp(\lambda = 1.44)$ | 1000 | 1 |
| J | 1000 | 8 | $\log(x) \sim N(12.8, 12.8)$ | $x \sim \exp(\lambda = 1.44)$ | 1000 | 1 |
| **Fig 4 panel** | | | | | | |
| A | | | $\log(x) \sim N(12.8, 12.8)$ | $x \sim \exp(\lambda = 1.1)$ | 1000 | 50 |
| B | | | $\log(x) \sim N(12.8, 12.8)$ | $x \sim \exp(\lambda = 1.1)$ | 1000 | 50 |
| D | | 8 | $\log(x) \sim N(12.8, 12.8)$ | $x \sim \exp(\lambda = 1.3)$ | 1000 | 20 |
| E | 1000 | 8 | $\log(x) \sim N(12.8, 12.8)$ | $x \sim \exp(\lambda = 1.3)$ | 1000 | 20 |
| F | 1000 | 8 | $\log(x) \sim N(12.8, 12.8)$ | $x \sim \exp(\lambda = 1.3)$ | 1000 | 20 |
| G | | 8 | $\log(x) \sim N(12.8, 12.8)$ | | 1000 | 50 |
| H | 100 | | $\log(x) \sim N(12.8, 12.8)$ | | 1000 | 50 |
| **Fig 5 panel** | | | | | | |
| A | | 8 | $\log(x) \sim N(12.8, 12.8)$ | | 1000 | 50 |
| B | | 8 | $\log(x) \sim N(12.8, 12.8)$ | | 1000 | 50 |
| C | 1000 | 8 | $\log(x) \sim N(12.8, 12.8)$ | | 1000 | 50 |

Detailed table of parameters used in all simulations. Note that $N(a, b)$ implies a Gaussian distribution with mean $a$ and variance $b$. Note $a = b$, based on [58,59].

The fixed observation period of $200$ $[ms]$ was divided into $K$ non-overlapping segments. During each segment $k \in [1, \ldots K]$, the response of neuron $i$ in population $j$ was drawn from a Poisson distribution. For the target neurons, the mean was $\frac{r_{i,j}}{K}$, whereas for distractor neurons, the mean was $\frac{r_{i,j}}{q_{i,j}K}$.

In each segment, the winning population was determined by the neuron with the highest spike count. The target's location was then estimated based on the receptive field of the population with the largest number of wins across all segments.

## Accuracy of WTA mechanisms computed directly from images

We expanded our investigation to assess the readout accuracy of WTA mechanisms; namely, the single-best-cell and the generalized population-based WTA, by incorporating results obtained from a complex structured model proposed by Itti et al. [6]. We utilized their provided code (http://ilab.usc.edu/bu/) without any alterations, ensuring the integrity of the methodology. The output of their code generated a '*Saliency Map*' with values ranging from $256$ to $0$ in arbitrary units. To generate the random neural responses to the stimulus we first translated the '*Saliency Map*' into firing rates. This was done by a linear transformation morphing the arbitrary units of the '*Saliency Map*' $256$ $(0)$ to firing rates of $12.8$ $[Hz]$ $(8.9$ $[Hz])$. In Fig 2A the neural responses were generated by Poisson statistics using these firing rates. In Fig 4G the neural responses were generated by Gaussian statistics using these firing rates as their mean responses. Our evaluations were conducted under conditions identical to those yielding our main results, as illustrated in both Fig 2A and Fig 4G marked by $x$'s. A comprehensive summary of the parameters is detailed in Table 1.

## Supporting information

**S1 Text.** ***Standard error of the mean of the readout accuracy in simulations.*** This file provides an analysis of the standard error of the mean (SEM) calculated over trials. It also presents (see $S1$ Fig in $S1$ text) the SEM, averaged over both trials and realizations, throughout the Results section.
(DOCX)

**S2 Text.** ***Discussion section: The error in estimating WTA and generalized WTA accuracy.*** This file shows (see $S2$ Fig in $S2$ text) the SEM of WTA and generalized WTA accuracy, as discussed in the Discussion section.
(DOCX)

## Acknowledgments

We are grateful to Mor Ben-Tov for helpful discussion about electrophysiological data of contextually modulated neurons in the optic tectum of the archerfish.

## Author contributions

**Conceptualization:** Ori Hendler, Maoz Shamir.

**Data curation:** Ronen Segev.

**Formal analysis:** Ori Hendler, Maoz Shamir.

**Funding acquisition:** Maoz Shamir.

**Investigation:** Ori Hendler, Ronen Segev, Maoz Shamir.

**Software:** Ori Hendler.

**Supervision:** Ronen Segev, Maoz Shamir.

**Validation:** Ronen Segev, Maoz Shamir.

**Visualization:** Ori Hendler.

**Writing – original draft:** Ori Hendler.

**Writing – review & editing:** Ronen Segev, Maoz Shamir.

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
