## [Decision Letter · Decision Letter 0]

12 Nov 2024

PCOMPBIOL-D-24-01308The Winner-take-all readout mechanism fails to account for the high accuracy reported in pop out visual searchPLOS Computational Biology Dear Dr. Hendler‬‏, Thank you for submitting your manuscript to PLOS Computational Biology. After careful consideration, we feel that it has merit but does not fully meet PLOS Computational Biology's publication criteria as it currently stands. Therefore, we invite you to submit a revised version of the manuscript that addresses the points raised during the review process.  As you will see from the comments below, the reviewers have some requests for changes with regards to potential further extensions to the model (if feasible, reviewer 2), and some rewriting and reorganisation of the sections (reviewer 3). Please notice that there are requirements for the sections in the manuscript, but it may still be possbile to move some descriptions of the model to the main text, in order to increase the readbility.  Please submit your revised manuscript within 60 days Jan 12 2025 11:59PM. If you will need more time than this to complete your revisions, please reply to this message or contact the journal office at ploscompbiol@plos.org. Please include the following items when submitting your revised manuscript: * A rebuttal letter that responds to each point raised by the editor and reviewer(s). You should upload this letter as a separate file labeled 'Response to Reviewers'. This file does not need to include responses to formatting updates and technical items listed in the 'Journal Requirements' section below.* A marked-up copy of your manuscript that highlights changes made to the original version. You should upload this as a separate file labeled 'Revised Manuscript with Track Changes'.* An unmarked version of your revised paper without tracked changes. You should upload this as a separate file labeled 'Manuscript'. If you would like to make changes to your financial disclosure, competing interests statement, or data availability statement, please make these updates within the submission form at the time of resubmission. Guidelines for resubmitting your figure files are available below the reviewer comments at the end of this letter. We look forward to receiving your revised manuscript. Kind regards, Ulrik R. BeierholmAcademic EditorPLOS Computational Biology Lyle GrahamSection EditorPLOS Computational Biology Feilim Mac GabhannEditor-in-ChiefPLOS Computational Biology Jason PapinEditor-in-ChiefPLOS Computational Biology  **Journal Requirements:** **Additional Editor Comments (if provided):****Reviewers' comments:** Reviewer's Responses to Questions

**Comments to the Authors:**

Reviewer #1: Visual search is a perceptual task in which a participant is instructed to locate an object of interest against a background of irrelevant objects. Of particular relevance to this manuscript is the finding that identifying a target amongst distractors that differ from the target by a unique visual feature is done automatically, with high speed and accuracy that are almost independent of the number of distractors. It is widely accepted that such “pop out” is achieved by winner-take-all (WTA) computation, and Hendler et al. use theoretical modeling to challenge this hypothesis. They convincingly show that heterogeneity in the neurons’ baseline firing rates critically limits the accuracy of such a model. The reason is that choices will be biased towards neurons whose baseline firing rate is relatively high.

The paper is interesting and insightful, and I have only few comments.

Relation to experiments:

The standard task in pop out tasks is to identify whether a unique object is presented. The answer to this question is yes/no and chance level is 50%. Indeed, in references (4) and (17), that are reproduced in Fig. 1D and 1E, respectively, these were the instructions of the participants. On the other hand, the question of the WTA algorithm is a different one: WHERE is the unique object? Hence chance level is reciprocal in the number of distractors. Therefore, using the latter to model the former is somewhat misleading. The authors should explain more precisely how their model can be mapped to the experimental results of references (4) and (17) or alternatively, discuss alternative experiments in which the task is to identify the location of the deviant.

Related to this point, the authors highlight the difference between pop out, and the serial nature of search when the unique object is characterized by a unique conjunction of features (Figs. 1C and 1D). However, it is not clear how the difference between these two modes of visual search manifest in the model.

A key component of the theory is the contextual modulation of the firing rate: the firing rate of a neuron to a stimulus presented in its receptive field depends also on the congruency of that stimulus with stimuli outside its receptive field. However, this is NOT what is presented in Fig. 1G, where we see the firing rate of a neuron when two different stimuli are presented to its receptive field (horizontal vs. vertical bar). This is certainly not what was done in reference 18 on which this figure is based. There, the stimulus in the receptive field was the same in the two conditions while the surround (context) varied. I guess that the illustration at the bottom of the figure is a typo. I suspect a similar typo related to Fig. 1H.

Clarity:

It is strange that Figure 1I is only discussed in the Methods section.

Line 79: the sentence “These studies… “ seems out of place

Lines 94-95: why is the time inherently independent of the number of distractors?

Line 191: The notation “ceil” in N_ceil and p_ceil is confusing, given the fact that in Fig 2C p_ceil can be 0.7, far from ceiling. I suggest changing the notations

Fig. 2: What is the number of distractors in A? Also, x-label is missing from A and B

Line 237-239: what scatter plot (x)?

Reviewer #2: This simulation study shows that two versions of the winner-takes all algorithm are unlikely to explain visual pop-out search. It makes an interesting point that is likely useful for future theorizing and empirical research; I could not discover any methodological flaws. From a cognitive perspective the basic idea seems a little naïve, though (but so does the highly successful salience model, this study is based on).

Signed. Heinrich Liesefeld

1. My only major critique of this study (which equally applies to the original Itti & Koch model, I believe) is that it is overly naïve from a cognitive perspective. Very early models of pop-out search already assumed that an initial attention allocation step that is informed by salience is followed by a verification step (Hoffman, 1979; see also Neisser, 1967 cited herein). The verification step is precisely introduced for the reason that basing a decision merely on the first (preattentive stage) is overly error prone. This basic assumption also made it into the currently most popular cognitive model of visual search (Guided Search, Wolfe, 1994, 2021). Another – somewhat less well known – solution to this problem is the repeated application of the same selection process (Humphreys & Müller, 1993, p. 59ff). I wonder how results would change if any of these slightly more sophisticated selection rules would be applied. For example, repeated application of the single-best-cell WTA would not suffer from the correlations in the generalized WTA and therefore yield similar results as the uncorrelated version (which made accurate predictions), I would think.

2. I am not sure I agree that divisive normalization can be relegated to future work. This is a highly influential idea in the context of salience/priority maps and if it had not been mentioned briefly in the Discussion (p. 29), I had brought it up in this review. This might just be my preference and for some other aspects, I would agree that these are beyond the scope of the present study, though. In general, some additional extensions (reducing the number of mechanisms that are “beyond the scope of the present study/work”) would certainly improve the potential impact of this paper.

Hoffman, J. E. (1979). A two-stage model of visual search. Perception & Psychophysics, 25(4), 319–327. https://doi.org/10.3758/BF03198811

Humphreys, G. W., & Muller, H. J. (1993). SEarch via Recursive Rejection (SERR): A connectionist model of visual search. Cognitive Psychology, 25(1), 43–110. https://doi.org/10.1006/cogp.1993.1002

Wolfe, J. M. (1994). Guided Search 2.0 A revised model of visual search. Psychonomic Bulletin & Review, 1(2), 202–238. https://doi.org/10.3758/BF03200774

Wolfe, J. M. (2021). Guided Search 6.0: An updated model of visual search. Psychonomic Bulletin & Review, 28, 1060–1092. https://doi.org/10.3758/s13423-020-01859-9

Reviewer #3: This is the review of "The winner-take-all readout mechanism fails to account for the high accuracy

reported in pop out visual search"

by Hendler, Segeve, Shamir, to Plos Computational Biology

This paper uses a modelling approach to argue that winner-take-all mechanisms cannot account for

the high accuracy in findinng out a pop-out item by humans and other animal species.

The model assumes a visual scene with a pop-out target item and many non-target items,

neurons respond to the target item and non-target items with different levels of

average responses, r_t and r_d, for the target and non-target respectively,

and their actual responses in a trial will be drawn from a probability distribution

whose averages will be r_t and r_d, with r_d = r_t/q by a q>1.

In an actual trial, the actual responses will not be the averages but

fluctuations around the averages, so that the highest response,

the winner response, from the population of neurons responding to the scene

may not be from a neuron responding to the target. Hence, the winner-take-all (WTA)

algorithm does not perform well compared to the actual performance of humans and

animals. The authors also considered an alternative, called

population-based WTA as opposed to single-best-cell WTA,

such that there are multiple neurons responding to each search item,

target or non-target, and the responses from the multiple neurons are averaged

to get an average response to each item, and the winner is then selected

from the winning average. They found that alternative also falls short.

In addition to pure modeling, the authors also used the neural data from archer

fish to give neural basis such as the value for q for their models.

The paper asks an interesting scientific question. However, there are

some serious problems with this paper that must be addressed.

(1) The paper aims to analyze the feasibility of the WTA readout by focusing on

issues such as number of distractors, population size, contextual modulation strength

(the value of q), heteogeneity of the neuronal population, and

noise correlations. In ths way, the paper builds various

probability distributions of the neural responses, and show

that WTA algorithm applied to the neural responses

could not account for the accuracies by human and animal data on pop-out tasks.

This conclusion is in contrast to that by Koene and Zhaoping 2007 and

Zhaoping and Zhe 2015. Using the WTA algorithm in the V1 Saliency Hypothesis,

and starting from human reaction times in pop-out search of some visual stimuli,

Zhaoping and colleagues quantitatively predict, with zero parameters, human reaction times

in pop-out search of some other visual stimuli, and these predictions match

observed data quantitatively.

The contrast between the conclusions of the current study and the previous works

suggest that this paper's models of probability distributions of the neural

responses (on which WTA must operation) may not be sufficiently

good models of the actual probability distribution of the neural responses.

Specifically, the probability distributions of the real

neural responses are most likely heavily influenced by recurrent interactions of

a neural circuit, such as the neural circuit in V1 (modeled by Li 1999,

also detailed in the textbook Zhaoping 2014), or the neural circuit in the optic

tectum of archer fish (see Zhaoping 2016 for a review of the evolution

of neural basis across species). The saliency mechanisms in these circuits

make neurons recurrently interact with each other via iso-feature suppression,

so that the neural responses to the pop-out target can be higher.

Such recurrent interactions in a neural circuit is likely to induce some kind of

fluctuations of the responses (for the target and non-target) that are not well

covered by the probability models in this paper. The introduction and discussion

of this paper should point out this limitation of the paper, and discuss

the relationship with previous works. Note that V1 as the neural basis for saliency

in primates has also strong neural data support, with simultaneous behavioral and neural

recordings from behaving monkeys doing a pop out task, by in Yan, Zhaoping, Li 2018.

(2) It is very difficult to read and understand the content of this

paper, making it difficult to review the paper. There are two problems

in the writing. First, many of the scientific and logical arguments are delegated as

mathematical derivations to Method section rather than having them in

the main text. This is not appropriate, since these scientific and logical arguments

are the main (rather than supplementary) content of this paper, which is after all a computational

paper. Second, perhaps because they are in the Methods section, they are not written

in a comprehensible way, making it difficult for a reviewer

to judge scientifically whether the arguments are sound.

The most important content should be written in the main text,

in a way that is not only understandable by the reviewers, but also

understandable to general readers of the PLoS computational biology.

I suggest that the authors test the paper to naive readers (outside

their lab) to see whether the paper is comprehensible in scientific arguments and

technical details. I should be able to more appropriately review this paper after

a revision is made to address this readability problem.

Below I also point out some other problems of this paper, some

to give specific examples where the writing is difficult

to read (I do not list all instances, just examples, so as

to give the authors an idea why they are difficult, to better

revise the paper), others to help with some important and

relevant references.

(3) Near line 100-103, references to Li 1999 PNAS should be added, it is

the first journal paper to explicitly propose a WTA algorithm on

V1 responses for saliency pop-out.

(4) The introduction can be better organized. By line 118, I was

wondering why "The goal of the study report below was to examine whether WTA

consittitutes the mechanism underling the pop out in the visual system"

is a modeling question rather than an empirical question. Only later

did I see what the goal was. This can be improved.

(5) Figure 2 some plots do not have axes marked, difficult for readers.

(6) Equation (6), what is i, i', j, j'? The index i' appears in the

right side but not the left side of the equation.

(7) The statement in line 484-486 has no reference!

(8) The paper says that rate model neurons are used, is this clear

whether the probabilities distributions are for rate neurons

or spiking neurons?

(9) In line 667, Kullback-Leibler divergence --- divergence between what?

**Have the authors made all data and (if applicable) computational code underlying the findings in their manuscript fully available?**

Reviewer #1: Yes

Reviewer #2: Yes

Reviewer #3: **No: ** as long as the paper is understandable, program codes are not critical

PLOS authors have the option to publish the peer review history of their article (what does this mean? ). If published, this will include your full peer review and any attached files.

**Do you want your identity to be public for this peer review?** For information about this choice, including consent withdrawal, please see our Privacy Policy .

Reviewer #1: No

Reviewer #2: **Yes: ** Heinrich R. Liesefeld

Reviewer #3: No

 **Figure resubmission:**While revising your submission, please upload your figure files to the Preflight Analysis and Conversion Engine (PACE) digital diagnostic tool, https://pacev2.apexcovantage.com/. PACE helps ensure that figures meet PLOS requirements. To use PACE, you must first register as a user. Registration is free. Then, login and navigate to the UPLOAD tab, where you will find detailed instructions on how to use the tool. If you encounter any issues or have any questions when using PACE, please email PLOS at figures@plos.org. Please note that Supporting Information files do not need this step. If there are other versions of figure files still present in your submission file inventory at resubmission, please replace them with the PACE-processed versions. 
---

## [Decision Letter · Decision Letter 1]

12 Mar 2025

PCOMPBIOL-D-24-01308R1

The winner-take-all readout mechanism fails to account for the high accuracy reported in pop out visual search

PLOS Computational Biology

Dear Dr. Hendler‬‏,

Thank you for submitting your manuscript to PLOS Computational Biology. After careful consideration, we feel that it has merit but does not fully meet PLOS Computational Biology's publication criteria as it currently stands. Therefore, we invite you to submit a revised version of the manuscript that addresses the points raised during the review process.

As you will see below, reviewer 3 still has some issues with the manuscript. Some of these relate to the clarity of the writing, and should be addressed. Others relate to the interpretation of the results, and should at least be discussed.

As a minor point, please do insert the figures into their proper places in the manuscript. Unfortunately the instructions for submission does not make it clear that this is possible, but it does make it easier to read the manuscript.

Please submit your revised manuscript within 30 days May 12 2025 11:59PM. If you will need more time than this to complete your revisions, please reply to this message or contact the journal office at ploscompbiol@plos.org. Please include the following items when submitting your revised manuscript:

We look forward to receiving your revised manuscript.

Kind regards,

Ulrik R. Beierholm

Academic Editor

PLOS Computational Biology

Lyle Graham

Section Editor

PLOS Computational Biology

**Journal Requirements:**

1) Some material included in your submission may be copyrighted. According to PLOSu2019s copyright policy, authors who use figures or other material (e.g., graphics, clipart, maps) from another author or copyright holder must demonstrate or obtain permission to publish this material under the Creative Commons Attribution 4.0 International (CC BY 4.0) License used by PLOS journals. Please closely review the details of PLOSu2019s copyright requirements here: PLOS Licenses and Copyright. If you need to request permissions from a copyright holder, you may use PLOS's Copyright Content Permission form.

Potential Copyright Issues:

i) Figures 1F. Please confirm whether you drew the images / clip-art (Human, Computer, and Torch) within the figure panels by hand. If you did not draw the images, please provide (a) a link to the source of the images or icons and their license / terms of use; or (b) written permission from the copyright holder to publish the images or icons under our CC BY 4.0 license. Alternatively, you may replace the images with open source alternatives. See these open source resources you may use to replace images / clip-art:

ii)  Figures 1D, 1E, 1G, 1H, and 2F. Thank you for stating that they have been adapted from previously published figures with permissions. Please provide copies of the permissions you have.

2) Please ensure that the funders and grant numbers match between the Financial Disclosure field and the Funding Information tab in your submission form. Note that the funders must be provided in the same order in both places as well. Currently, the order of the grants is different in both places.

**Reviewers' comments:**

Reviewer's Responses to Questions

Reviewer #1: The authors successfully addressed all my comments.

Reviewer #2: Even though – as also pointed out by Reviewer 3 – the present approach does not reflect the state-of-the-art in cognitive psychology, I already found the previous version of the manuscript of value for theorizing on salience computations. From my perspective, it helps to justify the increased complexity of empirically-informed cognitive theories from a computational standpoint and some steps were taken in this direction. Overall, I am happy with the way the authors responded to the minor concerns from my previous review and can therefore recommend the manuscript for publication.

Reviewer #3: This is the review of the revisiion of the paper "The winner-take-all readout mechanism fails to account for the

high accuracy reported in pop out visual search"

I appreciate that much of the mathematical derivations of this computational paper

is now moved into the main text. I also get to understand the paper's content better

than I did in the original version.

This paper uses a model of the visual responses x_i^j to a visual search item S_j,

and considers responses of many neurons to a visual search stimulus containing

a target t and many non-targets (or distractors ) d, and examines whether the

winner response ( i.e., the maximum x_i^j among responses with all possible i and j,)

is from a neuron responding to the target t in a pop-out search stimulus by a

sufficiently large probability (that reflects the high accuracy of human and animal's accuracy in

pop-out search). Hence, this model has two main components: (1) a model of the neural responses

to visual inputs, (2) a winner-take-all (WTA) algorithm to read out the winning response.

For (1), the paper assumes that x_i^j is a sample from a probability distribution

f(x_i^j |S_j), which is conditional to S_j (S_j = t or d), and this probability could be

conditionally independent between neurons x_i^j and x_{i'}^{j'}, or dependent on each other

by some correlation matrix C_{ii', jj'}.

The expectation value of x_i^j for S_t = t, i.e., E(x^j_i| S_j = t) and

that E(x^j_i| S_j = d) when S_j = d have a ratio q = E(x^j_i| S_j = t)/E(x^j_i| S_j = d).

The probability distribution f can be Poisson, exponential, etc. In addition,

the authors also used the Itti and Koch's saliency model to get the E(x^j_i| S_j = t) and

E(x^j_i| S_j = d) instead.

For (2), this study examined two WTA algorithms, one is the get the maximum response

among all x_i^j, the other is to average x_i^j across all i for a given j, and get

the winning average to identify the winning j.

The paper concludes on (2), the WTA algorithm, gives a poor accuracy to locate

the target. In other words, it often gives the winner, single neuron or average, as not one

responding to the target. This does not agree with the higher accuracy of pop-out search by humans and animals.

There is a main problem with this paper. The negative conclusion on (2), i.e., on WTA,

is conditioned on the model assumption on (1), i.e., on the model of the neural responses and their fluctuations.

The authors acknowledge this problem, in line 736-737, the paper states " ... the inherent tradeoff between

abstraction and biological realism emphasizes the need for caution in extrapolating our finding."

In particular, if (1) is grossly incorrect, then one should not attribute

the poor accuracy performance of the model to (2), i.e., to WTA. Hence, the title of the

paper is misleading as "The winner-take-all readout mechanism fails to account for the high

accuracy reported in pop out visual search". A reader seeing this title would assume

that WTA is the problem, when in fact the problem might be with the model of the

probability distributions of the neural responses to visual inputs. The authors

rightly pointed out that there are strong evidence for WTA in previous studies.

The title of this paper should be modified, perhaps to something like

"The winner-take-all readout mechanism fails to account for the high

accuracy reported in pop out visual search when neural responses to saliency are assumed as ... "

As pointed out in my last review, saliency neural responses are likely achieved by

a recurrent neural circuit, such as that in V1 in primates or optic tectum in archer fish

via the horizontal connections within a brain region. In particular, these horizontal

connections should make the response fluctuations in different neurons correlated

with each other by some kind of structure imposed by the recurrent interactions,

and they likely make the response to the target correlated with the response to

the distractors. Correlations in responses fluctuations could also arise from the arousal state of the animal.

These kinds of correlations are perhaps not according to the current model component (1) used by

the authors. Note that Itti and Koch's saliency model does not address this correlation in the

response fluctuations either, because that model does not use recurrent interactions to get the

responses to saliency. A biological plausible recurrent model for computing saliency is

Li (Zhaoping)'s model Li 1998 (which has all the model parameters listed,

the model parameters and many details are also available in the textbook Zhaoping 2014, if the authors

like to try it). That model uses noises in the neural dynamics (and noise induces response fluctuations),

but the saliency responses are robust and consistent with behavior. This paper should discuss this

issue as one of the main discussion points. To what degree the response fluctuations in a

recurrent neural circuit for saliency are related to the "Shared correlation" (shared in a particular way)

in this paper can be an interesting link for the discussion. This paper does recognize this issue of

"correlation in response fluctuations", as it states in line 644- 647 " If future electrophysiological studies

provide new evidence that can challenge our basic assumptions on the statistics

of neural responses, this will greatly expand ...". However, not addressing this

issue more directly, when this issue looms so large, could undermine the contribution

of this paper, as this paper could be viewed as a contribution to point out that

simple models of response fluctuations (unlike that induced in a recurrent circuit)

fail to account for pop-out saliency. Hence, one may say that the title of the paper

could also be revised to "Simple models of response fluctuations to saliency fails to account

for the high accuracy reported in pop out visual search when winner-take-all readout for saliency is used".

The authors could decide which title best highlights their contribution.

Additional comments:

There is still much room to improve the writing. E.g., in equation (2) , F_t, and F_d appear

without any definition. This is just an example, better writing could save reviewer/reader

a lot of time.

In the two sentences right after equation (9), maybe there is a typo, c1 and c2 should be swapped.

Line 479, neurons with closer tuning properties are more correlated --- this does

not necessarily mean that c1 >= c2. E.g., neurons tuned to the same orientations

can have very different receptive field locations.

Line 628, "... what are often modelled by saliency maps [96-99]", please note that

the references 98 and 99 are not models of the saliency maps, but are experimental

evidence supporting the idea that the saliency map is in V1, using WTA readout.

Another strong piece of evidence, perhaps visually more direct, for the WTA is Zhaoping

and May 2007 (Psychophysical tests of

the hypothesis of a bottom-up saliency map in primary visual cortex ), see their Fig. 1,

it suggests that WTA is not done by the winner of the local population

average (like a summation model by Itti and Koch), but winners among single neurons!

In figures, hard to distinguish between curves of different color. Having figures at the

end of the paper (rather than with the text), away from the figure legends, without clear legends in the plots

to denote which curves are for what (so that I do not understand unless I find and read the figure legends

somewhere else), makes this paper time consuming for reviewers to

read and review. For example, Figure 2, A, B, D, E, what are thered, blue, and black curves for?

**Have the authors made all data and (if applicable) computational code underlying the findings in their manuscript fully available?**

Reviewer #1: Yes

Reviewer #2: Yes

Reviewer #3: None

PLOS authors have the option to publish the peer review history of their article (what does this mean? ). If published, this will include your full peer review and any attached files.

**Do you want your identity to be public for this peer review?** For information about this choice, including consent withdrawal, please see our Privacy Policy .

Reviewer #1: **Yes: ** Yonatan Loewenstein

Reviewer #2: **Yes: ** Heinrich R. Liesefeld

Reviewer #3: No

**Figure resubmission:**
---

## [Editor Report · Decision Letter 2]

24 Apr 2025

Dear Mr. Hendler‬‏,

We are pleased to inform you that your manuscript 'Noise correlations and neuronal diversity may limit the utility of winner-take-all readout in a pop out visual search task' has been provisionally accepted for publication in PLOS Computational Biology.

Best regards,

Ulrik R. Beierholm

Academic Editor

PLOS Computational Biology

Lyle Graham

Section Editor

PLOS Computational Biology

---

## [Editor Report · Acceptance letter]

PCOMPBIOL-D-24-01308R2

Noise correlations and neuronal diversity may limit the utility of winner-take-all readout in a pop out visual search task

Dear Dr Hendler‬‏,

I am pleased to inform you that your manuscript has been formally accepted for publication in PLOS Computational Biology. Your manuscript is now with our production department and you will be notified of the publication date in due course.

With kind regards,

Zsofia Freund
